EMBO
Molecular Medicine

# Higher CSF sTREM2 and microglia activation are associated with slower rates of beta-amyloid accumulation

Michael Ewers[1,2,*] , Gloria Biechele[3], Marc Suárez-Calvet[4,5,6] , Christian Sacher[3], Tanja Blume[2], Estrella Morenas-Rodriguez[7], Yuetiva Deming[8] , Laura Piccio[9,10,11], Carlos Cruchaga[10,12], Gernot Kleinberger[7,13] , Leslie Shaw[14], John Q Trojanowski[15], Jochen Herms[2] , Martin Dichgans[1,2,16], the Alzheimer's Disease Neuroimaging Initiative (ADNI)[†], Matthias Brendel[3], Christian Haass[2,7,16] & Nicolai Franzmeier[1]

## Abstract

Microglia activation is the brain's major immune response to amyloid plaques in Alzheimer's disease (AD). Both cerebrospinal fluid (CSF) levels of soluble TREM2 (sTREM2), a biomarker of microglia activation, and microglia PET are increased in AD; however, whether an increase in these biomarkers is associated with reduced amyloid-beta (Aβ) accumulation remains unclear. To address this question, we pursued a two-pronged translational approach. Firstly, in non-demented and demented individuals, we tested CSF sTREM2 at baseline to predict (i) amyloid PET changes over ∼2 years and (ii) tau PET cross-sectionally assessed in a subset of patients. We found higher CSF sTREM2 associated with attenuated amyloid PET increase and lower tau PET. Secondly, in the $App^{NL-G-F}$ mouse model of amyloidosis, we studied baseline [18]F-GE180 microglia PET and longitudinal amyloid PET to test the microglia vs. Aβ association, without any confounding co-pathologies often present in AD patients. Higher microglia PET at age 5 months was associated with a slower amyloid PET increase between ages 5-to-10 months. In conclusion, higher microglia activation as determined by CSF sTREM2 or microglia PET shows protective effects on subsequent amyloid accumulation.

**Keywords** beta-amyloid accumulation; microglia; protective; tau; TREM2
**Subject Categories** Biomarkers; Neuroscience

## Introduction

The deposition of amyloid-β (Aβ) peptide is a key primary alteration that defines Alzheimer's disease (AD), involving a cascade of pathological brain changes resulting in the occurrence of dementia symptoms (Hardy & Higgins, 1992; Selkoe & Hardy, 2016). Recent evidence from genome-wide association studies suggests that alterations in the innate immune system are associated with enhanced risk of AD and thus modulate the development of AD (Lambert *et al*, 2009; Guerreiro *et al*, 2013; Jonsson *et al*, 2013; Reitz *et al*, 2013; Efthymiou & Goate, 2017; Sims

1 Institute for Stroke and Dementia Research (ISD), University Hospital, Ludwig Maximilian University (LMU), Munich, Germany
2 German Center for Neurodegenerative Diseases (DZNE), Munich, Germany
3 Department of Nuclear Medicine, University Hospital Munich, Ludwig Maximilian University Munich, Munich, Germany
4 Barcelonaβeta Brain Research Center (BBRC), Pasqual Maragall Foundation, Barcelona, Spain
5 IMIM (Hospital del Mar Medical Research Institute), Barcelona, Spain
6 Servei de Neurologia, Hospital del Mar, Barcelona, Spain
7 Chair of Metabolic Biochemistry, Biomedical Center (BMC), Faculty of Medicine, Ludwig-Maximilians-Universität München, Munich, Germany
8 Department of Population Health Sciences, University of Wisconsin-Madison, Madison, WI, USA
9 Department of Neurology, Washington University School of Medicine, St Louis, MO, USA
10 Hope Center for Neurological Disorders, Washington University School of Medicine, St. Louis, MO, USA
11 Brain and Mind Centre, University of Sydney, Sydney, NSW, Australia
12 Department of Psychiatry, Washington University School of Medicine, St. Louis, MO, USA
13 ISAR Bioscience GmbH, Planegg, Germany
14 Department of Pathology and Laboratory Medicine, Perelman School of Medicine, University of Pennsylvania, Philadelphia, PA, USA
15 Center for Neurodegenerative Disease Research, Institute on Aging, Perelman School of Medicine, University of Pennsylvania, Philadelphia, PA, USA
16 Munich Cluster for Systems Neurology (SyNergy), Munich, Germany
*Corresponding author. Tel: +49 89 4400 46221; E-mail: Michael.Ewers@med.uni-muenchen.de
†Data used in preparation of this article were obtained from the Alzheimer's Disease Neuroimaging Initiative (ADNI) database (adni.loni.usc.edu). As such, the investigators within the ADNI contributed to the design and implementation of ADNI and/or provided data but did not participate in analysis or writing of this report. A complete listing of ADNI investigators can be found in the appendix ("ADNI_coinvestigators.docx").

*et al*, 2017; Jansen *et al*, 2019). Microglia activation is the brain's major innate immune response to pathogens, and reactive microglia surrounding the amyloid plaques are a hallmark of AD (Itagaki *et al*, 1989). Microglia activation involves the phagocytosis of amyloid plaques as well as a reduction in Aβ-neurotoxicity and thus plays a key role to restore brain homeostasis (Sarlus & Heneka, 2017; Butovsky & Weiner, 2018). However, microglia activation can also be detrimental by inducing neurotoxic inflammation (Meda *et al*, 1995; Hong *et al*, 2016). Hence, understanding the role of microglia activation in response to core AD pathology (including Aβ) is pivotal for both tracking disease progression and identifying potential therapeutic interventions.

The overall aim of our study was to assess whether microglial function is associated with lower rates of Aβ accumulation throughout the Alzheimer's continuum. We addressed this question using two different approaches. First, in humans, we tested whether cerebrospinal fluid levels of triggering receptor expressed on myeloid cell type 2 (TREM2) as a marker of TREM2-mediated microglia response predict a change in accumulation of Aβ pathology. Second, we tested in a mouse model of amyloidosis whether changes in GE 180 (i.e., a PET tracer of microglial activation) predict longitudinal changes in Aβ deposition.

Our study on TREM2 was motivated by previous findings in both humans and animal models of Aβ, suggesting that TREM2-related microglia activation may have a protective role in AD (for review, see Butovsky & Weiner, 2018). TREM2 is a transmembrane protein of the immunoglobulin family that is expressed by immune cells throughout the body, but in the brain almost exclusively by microglia (Hickman *et al*, 2013). TREM2 forms a signaling complex with the DNAX activation protein 12 (DAP12) that triggers microglial activation, associated with increased microglial proliferation, chemotaxis, and phagocytosis of Aβ fibrils, as well as reduction of excessive pro-inflammatory responses (Takahashi *et al*, 2005; Wang *et al*, 2015; Zhong *et al*, 2015; Keren-Shaul *et al*, 2017; Mazaheri *et al*, 2017; Zhao *et al*, 2018). Genetic studies in humans showed that rare variants in the *TREM2* gene are associated with an exceptionally increased risk of AD, comparable to increase in risk by the *APOE ε4* genotype (Guerreiro & Hardy, 2013; Jonsson *et al*, 2013), suggesting that TREM2-related microglia activation may reduce the risk of AD dementia.

The soluble TREM2 (sTREM2) fraction is detectable in the cerebrospinal fluid (CSF) (Piccio *et al*, 2008), possibly through a mechanism of proteolytic cleavage of TREM2 and release into interstitial fluid (Kleinberger *et al*, 2014; Schlepckow *et al*, 2017). CSF sTREM2 is increased in humans with a biomarker profile of AD years before the onset of dementia symptoms (Heslegrave *et al*, 2016; Piccio *et al*, 2016; Suarez-Calvet *et al*, 2016a,b, 2019), where higher CSF sTREM2 at a given biomarker level of Aβ and pathologic tau was associated with larger gray matter volume (Gispert *et al*, 2016b) and slower subsequent cognitive and clinical decline in symptomatic elderly participants (Ewers *et al*, 2019). Together, these findings support a protective role of elevated sTREM2 in AD in the symptomatic phase of AD. However, whether increased CSF sTREM2 is associated with reduced rates of longitudinal increase of fibrillar Aβ is unclear. Therefore, in the first part of our study we set out to determine whether in elderly participants throughout the AD continuum,

CSF sTREM2 levels at baseline are associated with longitudinal changes in amyloid PET. In patients with AD, the rate of amyloid PET accumulation does not increase in a linear fashion, but shows a peak at about intermediate levels of global amyloid deposition before leveling off again, thus resembling a quadratic curve across the range of Aβ brain levels (Villemagne *et al*, 2013). Here, we hypothesized that higher CSF sTREM2 levels modulate this dynamic change in the rate of amyloid PET increase, i.e., higher CSF sTREM2 is associated with slower amyloid PET increase during the peak phase of amyloid deposition. A secondary goal in the current human PET study was to assess whether higher CSF sTREM2 levels are associated with lower tau PET independent of Aβ levels. Recent studies on mouse models of tau pathology suggest that that TREM2 deficiency is associated with higher tau pathology (Jiang *et al*, 2015, 2016), although inconsistent findings have been reported as well (Leyns *et al*, 2017). However, no study in humans has yet tested whether higher CSF sTREM2 is associated with lower tau PET in elderly subjects with biomarker evidence of Aβ.

In the last part of our study, we assessed microglia PET as a predictor of amyloid PET accumulation in a mouse model of genetically caused Aβ pathology. The use of mouse models of Aβ has the major advantage to assess any association between higher microglia activation and lower Aβ accumulation in a more controlled setting, i.e., in the absence of potentially confounding co-pathologies such as vascular changes that may drive microglia activation in patients with AD, where mixed pathologies are frequent (Kapasi *et al*, 2017). Furthermore, although CSF sTREM2 levels may reflect the amount of signaling-competent TREM2 on the surface of microglia (Kleinberger *et al*, 2014; Schlepckow *et al*, 2017) and thus provide a marker of microglia activation, the link between CSF sTREM2 and microglia activation is not firmly established. Therefore, we tested in mice the translocator protein-18 kDa (TSPO) PET as a marker of microglia activation in our second part of the study, reasoning that neuroimaging approaches including TSPO-PET allow for a more direct *in vivo* assessment of activated microglia activation in a longitudinal fashion (Edison *et al*, 2018). In longitudinal dual tracer studies, we previously showed that both microglia PET and amyloid PET are increased in an age-dependent manner in mouse models of amyloidosis (Brendel *et al*, 2017; Blume *et al*, 2018). Levels of sTREM2 in brain homogenate were associated with higher microglia PET in mouse models for amyloid pathology, which was positively correlated with higher immunohistochemically determined microglia numbers (Sacher *et al*, 2019), suggesting that higher microglia PET in mice with amyloidosis reflects higher microglia activation that is strongly correlated with TREM2 levels. We used this dual tracer paradigm in transgenic mice to test whether sporadically occurring interindividual variation in microglia activation at baseline is predictive of the rate of subsequent Aβ deposition. Specifically, we obtained both [18]F GE-180 TSPO microglia PET and [18]F-florbetaben amyloid PET in the *App*[NL-G-F] mice, which show global Aβ deposition at 4 months of age (Saito *et al*, 2014), reaching a plateau by 9 months of age (Mehla *et al*, 2019). We hypothesized that higher baseline microglia PET levels assessed at 5 months of age are associated with slower rates of amyloid PET increase during the subsequent follow-up period until 10 months of age.

# Results

## Patient characteristics

The baseline characteristics of all study participants are displayed in Table 1. The overall sample included 300 participants (136 female) with a mean (SD) age of 72.54 (6.4). The total sample included 94 CN Aβ− vs. 206 participants covering the Alzheimer's continuum (i.e., 55 CN Aβ+, 136 MCI Aβ+, 15 AD dementia). The sample size of subject in the AD continuum was sufficient to detect an effect size of $R^2 = 0.038$ for our hypothesized effect of the interaction CSF sTREM2 × AV45 PET on the rate of AV45 PET change at a power of 0.8 and alpha = 0.05. Compared to the CN Aβ− group, the Alzheimer's *continuum* group showed abnormal AD biomarker levels (i.e., AV45 global SUVR, CSF total tau, CSF p-tau$_{181}$) and cognition (ADNI-MEM, MMSE), and these abnormalities increased in symptomatic stages (i.e., MCI Aβ+, AD dementia). All group differences were assessed at the pre-defined alpha level of 0.05 (two-tailed).

## Inverse U-shaped association between baseline amyloid PET levels and annual amyloid accumulation

Previous studies in autosomal dominant and sporadic AD have shown that the rate of amyloid accumulation across the Alzheimer's *continuum* shows an inverse u-shape, where amyloid PET accelerates until a plateau is reached, after which amyloid PET accumulation decelerates (Villemagne *et al*, 2013; Gordon *et al*, 2018; Leal *et al*, 2018). To confirm this inversely u-shaped amyloid PET accumulation in the current sample, we assessed association between baseline AV45 PET levels and annual AV45 PET changes across the pooled sample (i.e., CN Aβ− & *Alzheimer's continuum*). Using a linear mixed effects model, we confirmed in our sample the hypothesized inversely u-shaped (i.e., quadratic) effect of baseline AV45 PET on annual AV45 PET change ($t(396) = -4.605$, $b/SE = -0.195/0.042$, 95% CI = $[-0.277; -0.113]$, $P < 0.001$, Cohen's $d = -0.463$, Fig 1A). Specifically, very low and very high baseline AV45 PET levels were associated with little annual AV45 PET changes, whereas intermediate AV45 PET levels were associated with relatively strong annual AV45 PET increases (controlling for baseline age, gender, education, diagnosis, CSF p-tau$_{181}$, *APOE ε4*, time between AV45 PET visits, and random intercept). The quadratic model of baseline AV45 PET on annual AV45 PET showed a better fit than the lineal model ($\chi^2(1) = 21.23$, $P < 0.001$). Testing the above described model in the Alzheimer's *continuum* group only ($t(271) = -3.354$, $b/SE = -0.253/0.076$, 95% CI = $[-0.399; -0.011]$, $P < 0.001$, Cohen's $d = -0.407$) and in symptomatic AD only (i.e., MCI and AD dementia, $t(226) = -2.240$, $b/SE = -0.185/0.083$, 95% CI = $[-0.344; -0.026]$, $P = 0.026$, Cohen's $d = -0.298$) yielded consistent results. This finding confirms previous evidence that annual AV45 PET increase (reflecting amyloid accumulation) accelerates initially but saturates at high AV45 PET levels. Results are summarized in Table 2, and all reported *P*-values are two-tailed.

## Higher CSF sTREM2 is associated with reduced amyloid PET accumulation

We next tested our major hypothesis that elevated CSF sTREM2 levels are associated with slower amyloid accumulation (i.e., an attenuated u-shaped association between baseline amyloid and amyloid accumulation). To this end, we tested whether CSF sTREM2 (i) had a main effect on annual AV45 PET changes or (ii) moderated the quadratic association between baseline AV45 PET levels and annual AV45 PET changes. We tested this using linear mixed effects models, controlling for baseline age, gender, education, diagnosis, CSF p-tau$_{181}$, *APOE ε4*, time between AV45 PET visits, and random intercept.

We found a significant main effect of CSF sTREM2 on annual AV45 PET change, such that individuals with higher CSF sTREM2 showed slower amyloid accumulation ($t(395) = -2.584$, $b/SE = -0.002/0.009$, 95% CI = $[-0.0041; -0.0005]$. $P = 0.010$, Cohen's $d = -0.260$). We further found a significant quadratic interaction CSF sTREM2 × baseline AV45 on annual AV45 changes, such that participants with higher CSF sTREM2 showed slower acceleration of AV45 PET increase than participants with low CSF sTREM2 levels ($t(342) = 2.460$, $b/SE = 0.211/0.086$, 95% CI = $[0.045; 0.376]$, $P = 0.014$, Cohen's $d = 0.266$, Fig 1B). Comparing the first model including the CSF sTREM2 main effect vs. the second model including the AV45 PET × CSF sTREM2 interaction revealed better model fit for the interaction model ($\chi^2(2) = 8.426$, $P = 0.015$). When testing this interaction model in the Alzheimer's *continuum* group only (with abnormally high AV45 PET baseline values), the interaction baseline AV45 PET × CSF sTREM2 on annual AV45 PET changes was pronounced ($t(268) = 3.861$, $b/SE = 0.636/0.165$, 95% CI = $[0.320; 0.952]$, $P < 0.001$, Cohen's $d = 0.472$). Consistent effects were found when further restricting the sample to symptomatic AD only, i.e., MCI Aβ+ & AD dementia ($t(225) = 3.225$, $b/SE = 0.601/0.186$, 95% CI = $[0.245; 0.957]$, $P = 0.001$, Cohen's $d = 0.432$). Results are summarized in Table 2, and reported *P*-values are two-tailed. In order to determine stages of amyloidosis in which beneficial sTREM2 effects on subsequent amyloid accumulation are largest, we performed sliding window analyses ($n = 100$ per window shifted in steps of 10) across low to high baseline AV45 levels. In each window, we assessed standardized group differences (i.e., Cohen's $d$) in subsequent AV45 change between subjects with high and low sTREM2 levels (i.e., defined via median split within each window). When plotting the resulting sTREM2-related differences in AV45 change, we found that beneficial effects of sTREM2 on future amyloid accumulation (i.e., positive Cohen's $d$ values) were highest (Cohen's $d \sim 0.2$–$0.4$) in windows around baseline AV45 SUVRs between 0.9–1.0, where annual AV45 change peaks (i.e., at baseline AV45 PET = 0.95, see Fig EV1). Together, these findings suggest that participants with higher CSF sTREM2 show slower amyloid accumulation than participants with lower CSF sTREM2.

## Higher GE-180 PET is associated with slower amyloid PET increase in an AD mouse model

We next tested whether the association between higher microglial activation and slower amyloid accumulation could be confirmed in a mouse model of amyloidosis. To this end, we assessed longitudinal Florbetaben PET (as a marker of Aβ accumulation) and GE-180 PET (as a marker of microglia activation) in transgenic *App$^{NL-G-F}$* mice.

First, we verified that the *App$^{NL-G-F}$* exhibited the expected increase in Florbetaben PET and GE-180 PET at baseline and over time when compared to that in the wild-type mice. Regression analyses showed

**Table 1. Baseline sample characteristics.**

| | Controls | | AD spectrum | | |
| --- | --- | --- | --- | --- | --- |
| | CN Aβ−<br>N = 94 | CN Aβ+<br>N = 55 | MCI Aβ+<br>N = 136 | AD dementia<br>N = 15 | P-value |
| Age | 72.9 (5.88)[d] | 72.9 (6.00)[d] | 71.6 (6.54)[d] | 77.9 (7.22)[a,b,c] | 0.003 |
| Gender (f/m) | 41/53 | 34/21[c] | 56/80[b] | 5/10 | 0.047 |
| Education | 16.9 (2.37) | 16.2 (2.67) | 16.2 (2.61) | 15.5 (2.70) | 0.099 |
| ApoE4 (pos/neg) | 81/13[b,c,d] | 30/25[a,d] | 51/85[a] | 2/13[a,b] | < 0.001 |
| ADNI-MEM | 1.18 (0.56)[c,d] | 1.17 (0.68)[c,d] | 0.18 (0.61)[a,b,d] | −0.83 (0.33)[a,b,c] | < 0.001 |
| MMSE | 29.2 (1.07)[c,d] | 29.1 (0.88)[c,d] | 27.8 (1.81)[a,b,d] | 22.5 (1.81)[a,b,c] | < 0.001 |
| AV45 global SUVR | 0.74 (0.03) | 0.91 (0.10) | 0.97 (0.11) | 1.05 (0.07) | < 0.001 |
| Time between AV45 visits | 2.16 (0.59)[b,c,d] | 2.33 (0.85)[a,c,d] | 2.12 (0.50)[a,b,d] | 2.00 (0.03)[a,b,c] | 0.098 |
| CSF sTREM2 | 4,163 (2,086) | 3,701 (2,007) | 4,317 (2,212) | 4,640 (2,599) | 0.270 |
| CSF total tau | 220 (78.3)[c,d] | 268 (107)[c,d] | 333 (132)[a,b] | 381 (118)[a,b] | < 0.001 |
| CSF p-tau$_{181}$ | 19.3 (7.11)[b,c,d] | 25.8 (11.4)[a,c,d] | 32.9 (14.4)[a,b] | 39.1 (12.9)[a,b] | < 0.001 |
| Number of participants with tau PET[e] | 14 | 14 | 26 | 0 | |

[a]Sig. (P < 0.05) different from CN AB−.
[b]Sig. (P < 0.05) different from CN AB+.
[c]Sig. (P < 0.05) different from MCI AB+.
[d]Sig. (P < 0.05) different from AD dementia.
[e]Tau PET had to be obtained maximum 3 years after the last joint AV45 PET/CSF sTREM2 assessment.

that at baseline (5 months), GE-180 PET ($t(56) = 8.369$, $P < 0.001$, Fig 2A) and Florbetaben PET ($t(56) = 2.652$, $P = 0.010$) were higher in $App^{NL-G-F}$ compared to age- and gender-matched C57BL/6 control mice, suggesting both amyloidosis and microglial activation in the AD mouse model. When assessing the rate of increase in Florbetaben PET at from 5 to 10 months of age, $App^{NL-G-F}$ mice also showed faster increase in Florbetaben PET compared to C57BL/6 control mice ($t (56) = 5.551$, $P < 0.001$, Fig 2B).

Next, we tested our main hypothesis, i.e., whether higher baseline GE-180 PET (assessed at 5 months) predicts slower rate of Florbetaben PET up to 10 months of age. Using linear regression, we found that higher baseline GE-180 PET levels (continuous measure) predicted slower longitudinal increase in Florbetaben PET in the $App^{NL-G-F}$ mice, controlling for baseline Florbetaben PET ($t (12) = −2.354$, $b$/SE $= −0.367/0.198$, $P = 0.036$, Fig 2C). Figure 3 illustrates the GE-180 and Florbetaben uptake in the C57BL/6 control mice (Fig 3A) and the $App^{NL-G-F}$ split into subgroups of low (Fig 3B) and high (Fig 3C) baseline GE-180 PET. Note that continuous rather than binarized GE 180 PET levels were included as a predictors in the regression analysis described above to avoid any potential bias. Visual inspection of the figure shows that $App^{NL-G-F}$ mice with high baseline microglia activity showed a reduced increase in amyloid PET uptake during the ensuing 5 months of follow-up compared to those $App^{NL-G-F}$ mice who showed low GE-180 PET at baseline, confirming that higher GE 180 PET at baseline is associated with reduced subsequent increase in Florbetaben PET.

**Higher sTREM2 is associated with lower tau PET independent of amyloid**

In a last step, we tested whether higher sTREM2 levels are associated with lower tau PET uptake. Since tau PET was only introduced

recently in ADNI3, all CSF sTREM2 measures were obtained prior to tau PET assessments. For the current analyses, we selected only 54 subjects for which tau PET was acquired maximum 3 years after a joint CSF sTREM2 & AV45 assessment (i.e., 14 CN Aβ−, 14 CN Aβ+, 26 MCI Aβ+). The mean time difference between CSF sTREM2/ AV45 and tau PET assessment was $1.6 \pm 1.12$ years. Using ANCOVAs, we found that subjects with higher sTREM2 levels showed lower tau PET SUVR in Braak-stage 1 ROIs (i.e., entorhinal cortex; $F_{7,46} = 9.527$, $P = 0.005$) & Braak-stage 3&4 ROIs ($F_{7,46} = 5.253$, $P = 0.032$), controlling for age, gender, education, diagnosis, ApoE4 status, and AV45. No significant effects of CSF sTREM2 were found, however, on tau PET SUVR in Braak-stage 5&6 ROIs ($F_{7,46} = 2.810$, $P = 0.108$). Results are summarized in Fig 4. When applying a Bonferroni-corrected alpha threshold of 0.0167, the effect of CSF sTREM2 on tau PET SUVR in Braak-stage 1 (i.e., entorhinal cortex) remained significant.

## Discussion

The primary finding of the current study was that higher CSF sTREM2 levels at baseline were associated with slower rates of Aβ accumulation as assessed by amyloid PET over 2 years in elderly participants, supporting a protective role of higher CSF sTREM2 in the development of amyloid pathology. Secondary analysis of the CSF sTREM2 effects on AV1451 tau PET showed that higher CSF sTREM2 was associated with lower tau PET uptake in early Braak-stage regions, supporting a protective role of TREM2 in the development of key AD pathology. The second major finding was that in a transgenic mouse model of Aβ pathology, higher microglia PET signal at baseline was associated with slower rate of amyloid PET increase. These results are supportive of our findings in humans.

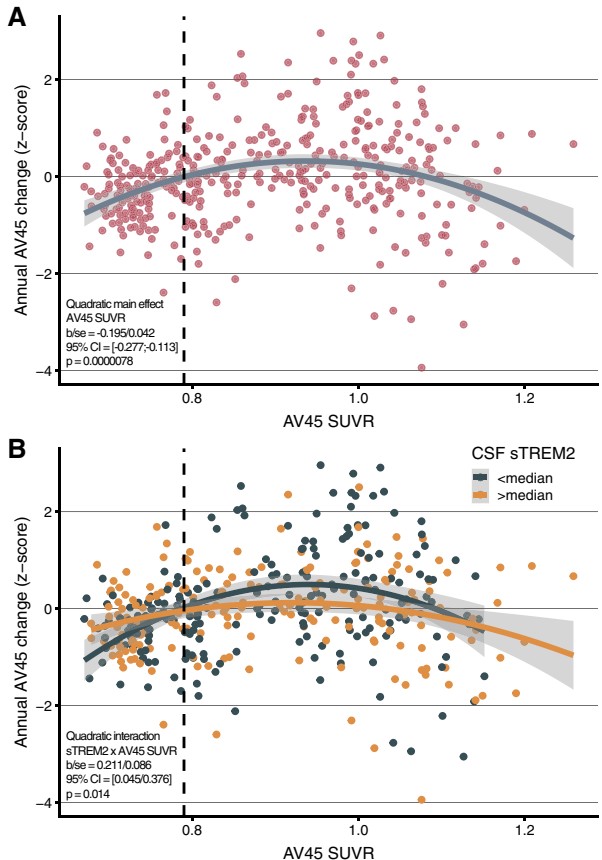

**Figure 1. Association between baseline amyloid and amyloid PET change.**

A, B   Regression plot showing the association between baseline AV45 PET (*x*-axis) and annual AV45 PET change (*y*-axis) across the entire sample (*N* = 300). The vertical dashed line represents the Aβ-positivity threshold of AV45 PET SUVR > 0.079 (A). Regression plot for the association between baseline AV45 PET and annual AV45 PET change stratified by baseline CSF sTREM2 levels (B).

Source data are available online for this figure.

Although the current observational study does not purvey a causative effect of sTREM2 or microglia activation on Aβ accumulation, our results support the view that microglia activation during intermediate levels of Aβ is associated with less strong Aβ accumulation.

Consistent with previous studies (Villemagne *et al*, 2013; Gordon *et al*, 2018; Leal *et al*, 2018), we found that the rate of Aβ accumulation follows a quadratic function across baseline Aβ levels, i.e., the increase accelerates until an intermediate level of Aβ deposition before gradually slowing down again. Importantly, we found that higher levels of CSF sTREM2 alter this dynamic change in Aβ, reducing the rate of increase in amyloid PET during the phase of the strongest increase in Aβ. Thus, we demonstrate here that CSF sTREM2 modulates the accumulation of a key AD pathology, extending previous reports of higher CSF sTREM2 (Gispert *et al*, 2016b; Ewers *et al*, 2019) and higher microglia PET (Hamelin *et al*, 2016) to be associated with lower brain atrophy and slower cognitive decline in patients with AD. Furthermore, a common genetic

variant in the 4-domain subfamily gene MS4A4A that is associated with *decreased* risk of AD was found to be associated with *higher* CSF sTREM2 levels (Deming *et al*, 2019), suggesting that higher sTREM2 levels may contribute to a decreased risk of AD. The current findings are also broadly consistent with a previous brain autopsy study reporting that individuals carrying *TREM2* mutations exhibit lower amyloid plaque-associated microglia activation, less microglial clustering (Parhizkar *et al*, 2019), and higher neuritic plaque score (Prokop *et al*, 2019). Neuritic plaques are the most important source for amyloid PET binding (Clark *et al*, 2012; Ikonomovic *et al*, 2016), and thus, the current findings of higher CSF sTREM2 to be associated with lower amyloid PET binding are consistent with the post-mortem findings of TREM2 risk variants and increased neuritic plaque score.

Our observation of higher CSF sTREM2 and lower amyloid PET is further supported by results from transgenic mouse models, where a number of studies suggest a protective effect of TREM2-related microglia activation on Aβ (Yuan *et al*, 2016; Parhizkar *et al*, 2019; for review, see Butovsky & Weiner, 2018). In 5×FAD mice which show Aβ deposition in the hippocampus and cortex by 6 months of age (Oakley *et al*, 2006), knockdown of the *TREM2* gene decreased microglia activation but increased amyloid plaque load at ages 8 months in the hippocampus (Wang *et al*, 2015) and cortex (Griciuc *et al*, 2019). In contrast, overexpression of the TREM2 gene induced lower amyloid plaque load at 7 months (Lee *et al*, 2018). In APP-PS1 mice, which show widespread amyloid plaques in the brain by 3–4 months of age (Radde *et al*, 2006), *TREM2* knockdown was associated with higher amyloid plaque burden at 8 months (Jay *et al*, 2017) and 12 months (Sheng *et al*, 2019), i.e., when amyloid plaque deposition is already quite expanded but still increasing. *TREM2* knockout in the APPPS1 mice was associated with increased Aβ seeding induced amyloid plaque burden at 6 months of age (Parhizkar *et al*, 2019). Together, these results suggest a protective effect of TREM2 on the accumulation of Aβ during a phase of intermediate levels of Aβ, i.e., when Aβ accumulation is still increasing over time. This observation in the transgenic mice is consistent with our results of CSF sTREM2 to modulate the increase in Aβ accumulation particularly during intermediate levels of Aβ (as shown by the interaction between sTREM2 and the quadratic term of AV45 PET). This stance is consistent with the proposal that any TREM2-dependent beneficial effects on amyloid are stage-dependent and occur preferentially at later stages of Aβ (Jay *et al*, 2017), which may explain why some studies did not find a beneficial effect of TREM2 on Aβ (Jay *et al*, 2015; Krasemann *et al*, 2017).

In addition to an association between higher CSF sTREM2 and lower rates of amyloid deposition, we found in a secondary analysis that higher CSF sTREM2 was associated with lower cross-sectionally assessed levels of AV1451 tau PET, especially in early Braak-stage ROIs. The time delay between CSF sTREM2 and the single AV1451 PET did not permit to test whether the effect of CSF sTREM2 on AV1451 PET was mediated by lower rates of amyloid PET. Interestingly, a recent study in a transgenic mouse model of Aβ reported that genetically induced TREM2 deficiency was associated with higher pathologic tau seeding in neurotic plaques (Leyns *et al*, 2019). These results correspond well to post-mortem histochemical findings of higher tau burden in the microenvironment of amyloid plaques in AD patients with the TREM2 mutations (Prokop *et al*,

**Table 2. Linear mixed models testing the effects of baseline AV45 and CSF sTREM2 on annual AV45 PET change.**

| Dependent variable: annual AV45 change | $b$/SE | 95% CI | $t$-value | $P$-value | Cohen's $d$ | Partial $R^2$ |
|---|---|---|---|---|---|---|
| **Whole group** | | | | | | |
| Model 1[a]: Main effect—Baseline AV45 SUVR$^2$ | −0.195/0.042 | −0.277; −0.113 | −4.605 | < 0.001 | −0.463 | 0.050 |
| Model 2[b]: Main effect—Baseline CSF sTREM2 | −0.023/0.009 | −0.0041; −0.0006 | −2.584 | 0.010 | −0.260 | 0.016 |
| Model 3[b]: Interaction—Baseline AV45 SUVR$^2$ × CSF sTREM2 | 0.211/0.086 | 0.045; 0.376 | 2.460 | 0.014 | 0.266 | 0.015 |
| **AD continuum** | | | | | | |
| Model 1[a]: Main effect—Baseline AV45 SUVR$^2$ | −0.253/0.076 | −0.399; −0.011 | −3.354 | < 0.001 | −0.407 | 0.039 |
| Model 2[b]: Main effect—Baseline CSF sTREM2 | −0.003/0.001 | −0.0053; −0.0006 | −2.400 | 0.017 | −0.292 | 0.020 |
| Model 3[b]: Interaction—Baseline AV45 SUVR$^2$ × CSF sTREM2 | 0.636/0.165 | 0.320; 0.952 | 3.861 | < 0.001 | 0.472 | 0.051 |
| AD clinical (MCI Aβ+ & AD Dementia)[a] | | | | | | |
| Model 1[a]: Main effect—Baseline AV45 SUVR$^2$ | −0.185/0.083 | −0.344; −0.026 | −2.240 | 0.026 | −0.298 | 0.021 |
| Model 2[b]: Main effect—Baseline CSF sTREM2 | −0.004/0.001 | −0.0062; 0.0009 | 2.563 | 0.011 | −0.342 | 0.027 |
| Model 3[b]: Interaction—Baseline AV45 SUVR$^2$ × CSF sTREM2 | 0.601/0.186 | 0.245; 0.957 | 3.225 | 0.001 | 0.432 | 0.042 |
| AD preclinical (CN Aβ+) | | | | | | |
| Model 1[c]: Main effect—Baseline AV45 SUVR$^2$ | −0.718/0.176 | −1.043; −0.392 | −4.092 | < 0.001 | −1.047 | 0.197 |
| Model 2[d]: Main effect—Baseline CSF sTREM2 | −0.002/0.002 | −0.006; 0.002 | −0.802 | 0.426 | −0.207 | 0.009 |
| Model 3[d]: Interaction—Baseline AV45 SUVR$^2$ × CSF sTREM2 | −0.214/0.364 | −0.872; 0.443 | −0.589 | 0.558 | −0.155 | 0.005 |

[a]Model controlled for age, gender, education, diagnosis, CSF p-tau$_{181}$, ApoE4 status, time between AV45 visits, and random intercept.
[b]Model controlled for AV45 SUVR$^2$, age, gender, education, diagnosis, CSF p-tau$_{181}$, ApoE4 status, time between AV45 visits, and random intercept.
[c]Model controlled for age, gender, education, CSF p-tau$_{181}$, ApoE4 status, time between AV45 visits, and random intercept.
[d]Model controlled for AV45 SUVR$^2$, age, gender, education, CSF p-tau$_{181}$, ApoE4 status, time between AV45 visits, and random intercept.
All $P$-values are based on two-tailed alpha thresholds.

2019). These results suggest the possibility that TREM2 plays an important role in Aβ-related tau pathology, which could potentially underlie the CSF-TREM2-associated decrease in neurofibrillary tangle deposition observed in the current study. However, whether a direct mechanistic link between TREM2 function and neurofibrillary tangles exists remains to be investigated.

For the association between higher CSF sTREM2 and lower amyloid PET accumulation, several potential mechanisms may account for the observed results. Causal gene expression network analysis showed the TREM2/DAP12 complex as a key regulator hub for microglia gene expression in AD (Zhang et al, 2013; Wang et al, 2015; Keren-Shaul et al, 2017; Parhizkar et al, 2019). TREM2 activation increases the ApoE-dependent phenotypic switch from a homeostatic to a disease-associated microglial signature in response to Aβ (Krasemann et al, 2017; Parhizkar et al, 2019). Furthermore, TREM2 deficiency causes reduced proliferation (Kleinberger et al, 2017; Zhao et al, 2018) and chemotaxis of microglia (Mazaheri et al, 2017), resulting in reduced number of microglia around amyloid plaques and phagocytosis of Aβ (Zhao et al, 2018; Parhizkar et al, 2019). Thus, TREM2 may be key to microglia activation with beneficial effects on Aβ pathology.

Our findings in the transgenic mouse model of Aβ pathology demonstrating higher microglia PET at baseline to be associated with slower amyloid PET accrual are also consistent with the idea of a protective role of microglia activation. Our results are in general agreement with previous reports of experimentally increased TREM2 to be associated with lower Aβ deposition (Lee et al, 2018), and genetically induced TREM2 deficiency to be associated with higher Aβ deposition at a time window of advanced but still accumulating Aβ in transgenic mouse models of Aβ (Jay et al, 2017;

Sheng et al, 2019). Importantly, our results suggest that lower amyloid rates were associated with higher *sporadically* occurring microglia activity, rather than as a consequence of microglia activation manipulated by genetic mutations, which in humans only rarely occurs in carriers of *TREM2* risk variants. Our findings provide thus an important advance because microglia alterations, such as those induced by loss-of-function *TREM2* risk variants, may not necessarily be translatable to the effects of increased microglia activation in disease. Our findings on beneficial associations of microglia PET with subsequent Aβ during a phase of advanced yet still increasing Aβ in the mouse model of Aβ show parallels to our findings on higher CSF sTREM2 to be associated with lower subsequent amyloid PET increase in elderly individuals, in particular during a phase of intermediate Aβ deposition at baseline. The results are also consistent with previous findings on [18F] DPA-714 TSPO-PET, a marker of microglia activation, in patients with prodromal AD and AD dementia, where higher levels of baseline microglia PET showed protective effects on the clinical progression (Hamelin et al, 2016, 2018). Overall, these results support that higher microglia activation modulates the accumulation of Aβ in both mice and humans, and can obtain a protective role.

The mechanisms underlying differences in microglia activation at the same level of pathology are unclear. One possible source is that microglia activation may differ as a function of different amyloid plaque morphologies (Griffin et al, 1995). Microglia activation is most strongly linked to neuritic rather than diffuse plaques (Sheng et al, 1997). Thus, it is possible that mice with higher GE 180 PET at baseline had more mature neuritic plaques and thus advanced levels of amyloid deposition, entailing a ceiling effect. However, it is

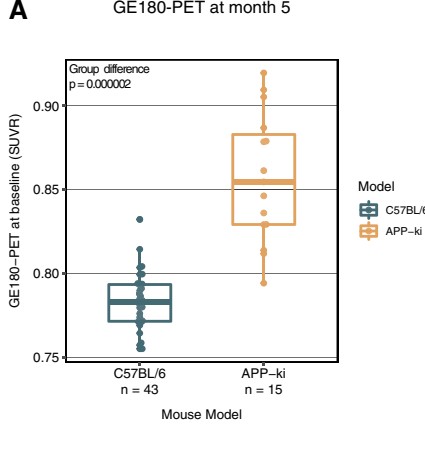

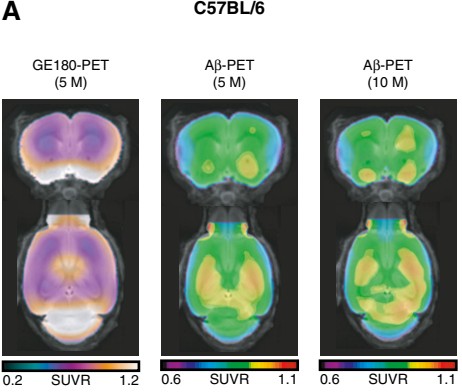

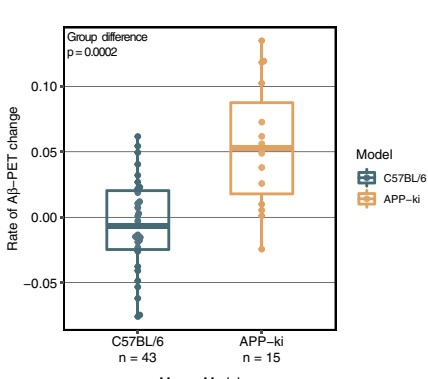

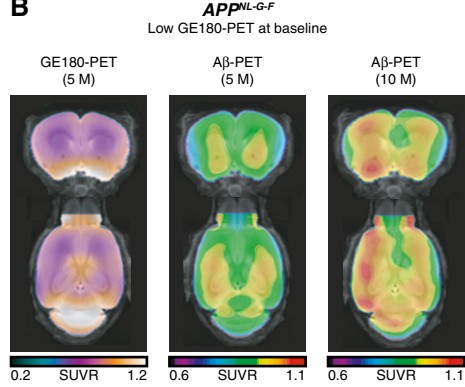

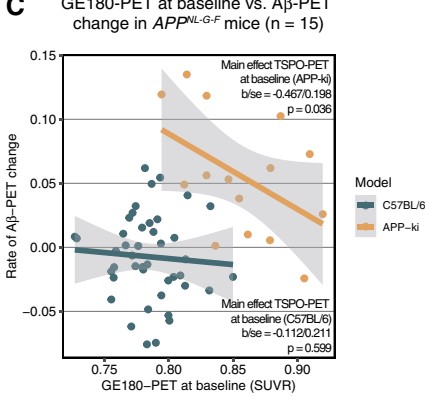

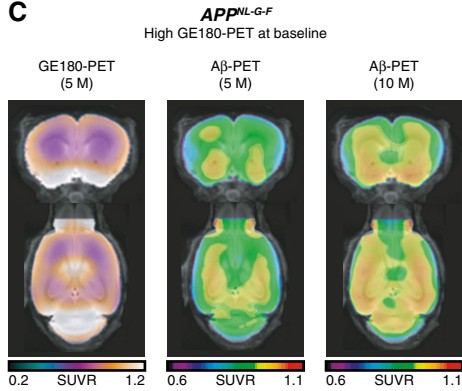

**Figure 2. Rate of change in Florbetaben PET vs. baseline GE180 PET.**

A, B Boxplots (central band = median, boxes = quartiles, whiskers = 1.5 * interquartile range), illustrating the differences in GE180 PET (A) and the rate of change in Florbetaben PET (B) compared between wild-type C57BL/6 mice (*n* = 43) and APP$^{NL-G-G}$ (*n* = 15).

C The regression plot shows lower rate of change in Florbetaben PET to be associated with lower GE180 PET at baseline, i.e., at 5 months of age (C). The shaded area corresponds to the 95% CI of the regression line.

Source data are available online for this figure.

**Figure 3. Brain rendering of GE180 and Florbetaben PET by group and time point.**

A–C Coronal (top row) and axial (bottom row) slices of average 18-F GE-180 TSPO (microglia) tracer and $^{18}$F-florbetaben amyloid PET split up by group (A–C): C57BL/6 (A), APP$^{NL-G-F}$ mice subgroup with a low (< median) GE-180 PET (B), and the APP$^{NL-G-F}$ mice subgroup with a high (> median) GE-180 PET (C). APP$^{NL-G-F}$ mice with low baseline GE180 PET levels showed faster increase in Florbetaben PET between 5 and 10 months (B) compared to those at high baseline level of GE-180 PET (C).

unlikely that ceiling effects of amyloid deposition explain the current results, because in agreement with previous studies (Mehla *et al*, 2019), the App$^{NL-G-F}$ mice in the current study showed

substantial increase in Florbetaben PET between 5 and 10 months of age. Another possibility is that other less controllable interindividual differences in behavior such as physical activity and exploratory behavior may have modulated microglia activity in the

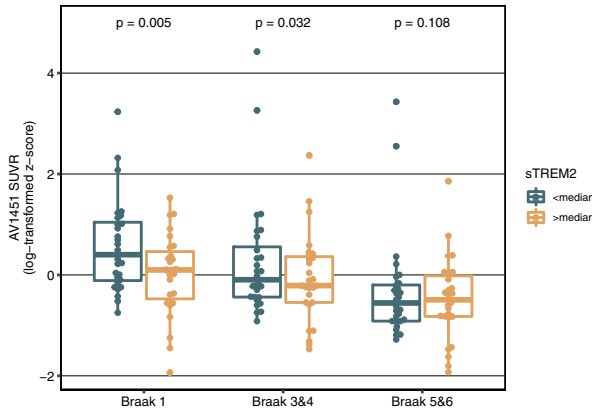

**Figure 4. TREM2 vs. tau PET.**
Effects of sTREM2 on AV1451-PET-assessed tau pathology stratified by Braak stages shown as boxplots (central band = median, boxes = quartiles, whiskers = 1.5 * interquartile range).
Source data are available online for this figure.

transgenic mice. A number of previous studies that systematically stimulated physical activity and exploratory behavior by enriched environment showed a reduction in the accumulation of soluble Aβ and amyloid plaque burden in transgenic mouse models of Aβ (Moore *et al*, 2016; Zhang *et al*, 2018), possibly mediated by enhanced microglial phagocytotic activity and thus Aβ deposition (Ziegler-Waldkirch *et al*, 2018). However, we did not investigate systematically physical or exploratory activities in the mice, and thus, the sources of sporadic differences in microglia activation at the same level of Aβ remain to be clarified in future studies.

Some caveats need to be taken into account for the interpretation of the current findings.

In humans, we assessed CSF levels of sTREM2 rather than the actual TREM2 or microglia activation in the brain. However, several lines of evidence support the notion that sTREM2 probably reflects the amount of signaling-competent TREM2 on the surface of activated microglia and can be thus used as a surrogate marker of TREM2-mediated microglia response: First, the levels of sTREM2 in the brain of an Aβ mouse model correlate with TSPO-PET signal (Brendel *et al*, 2017). Second, mouse models bearing the *TREM2* p.T66M mutation show decreased microglial activity (Kleinberger *et al*, 2017). Lastly, individuals with *TREM2* risk variants have reduced post-mortem levels of microglia surrounding amyloid plaques (Prokop *et al*, 2019), and CSF sTREM2 levels are altered in *TREM2* risk variants *in vivo* (Suarez-Calvet *et al*, 2019).

Another caveat is that measuring CSF sTREM2 does not allow us to differentiate between the effect of TREM2 or a potential unique role of its soluble form. Injection of sTREM2 into the brains of transgenic mice with Aβ deposition reduced amyloid plaque load in a microglia-mediated way (Zhong *et al*, 2019), suggesting that sTREM2 may exert a functional role in reducing Aβ rather than being an inert cleavage product. Still, several observations suggest that the current findings are unlikely to be reduced to direct effects of sTREM2 on Aβ. As mentioned above, sTREM2 in brain homogenate was associated with higher

microglia PET suggesting that sTREM2 varies as a function of microglia activity and TREM2 loss-of-function homozygous mutations are associated with dramatically decreased levels of sTREM2. Together, these results suggest that higher sTREM2 levels are associated with higher microglia activation. Still, we acknowledge that the association between CSF sTREM2 levels and microglia activation remains to be established in humans. Another caveat is that the current study is correlational in nature and does not test microglia activation as a causal event. However, our stepwise effect size analysis showed that the size of the association between high CSF sTREM2 and rate of amyloid PET change was highest around the peak rate of AV45 PET accumulation, speaking against the alternative hypothesis that high levels of CSF sTREM2 simply coincide with low amyloid accumulation during the course of AD. Still, it is important to note that the current study does not necessarily imply that microglia activation causes lower amyloid accumulation, rather we demonstrate that CSF sTREM2 levels and microglia PET are predictive of lower future amyloid PET accumulation. Furthermore, microglia activation and neuroinflammation in AD are complex and heterogenous involving multiple pathways (e.g., IL6, IL-1β, and YKL-40) and several studies suggest that microglia activation can have adverse effects in AD (Llorens *et al*, 2017; Kinney *et al*, 2018; Gotzl *et al*, 2019; for review, see Butovsky & Weiner, 2018). For example, CSF levels of YKL-40 have been previously reported to be increased in Alzheimer's disease, peaking at the prodromal stage (Gispert *et al*, 2016a), similarly to what has been described for CSF sTREM2 (Suarez-Calvet *et al*, 2016b). It remains to be investigated whether YKL-40 regulates the rate of amyloid pathology, which we could not address in the current study. The current study is limited with regard to range of AD disease severity, but effects of microglia activation may vary by disease stage as suggested by results from transgenic mice (Jay *et al*, 2017; Parhizkar *et al*, 2019; Meilandt *et al*, 2020) and clinical trials in humans (Leoutsakos *et al*, 2012). Furthermore, we focus mainly on Aβ whereas the evidence for sTREM2 effects on pathologic tau remains preliminary and may be differentially affected by microglia activation (Bemiller *et al*, 2017; Leyns *et al*, 2017; Jiang *et al*, 2018). For findings in the App[NL-G-F] and C57BL/6 control mice, a substantial variability of TSPO expression and rate of change in Florbetaben PET was observed. However, previous studies demonstrated for both PET tracers a high correlation between PET uptake and the measurements obtained by the corresponding immunohistochemical gold standard methods of amyloid deposition or microglia activity (Rominger *et al*, 2013; Brendel *et al*, 2015, 2016; Parhizkar *et al*, 2019; Sacher *et al*, 2019). That variability likely represents a mixture between methodological variability and interanimal heterogeneity, potentially stemming from interindividual differences in the level of experienced stress (Calcia *et al*, 2016), physical activity (Kohman *et al*, 2013), or different stages of the menstrual cycle (Thakkar *et al*, 2018). However, these parameters were not assessed in the current study and thus such potential explanations remain to be confirmed.

Overall, our findings suggest that microglia activation may play a protective role in the reduction of Aβ in AD. These results have direct therapeutic relevance. Stimulation of microglia activation such as through TREM2 antibodies is currently being tested in phase 1 trials in AD (https://clinicaltrials.gov/ct2/show/NCT03635047). The tracking of microglia activity via CSF sTREM2 levels, microglia

PET, and amyloid PET will play a pivotal role in the evaluation of the effectiveness of such interventions in AD.

# Materials and Methods

### Participants

A total of 300 participants were included from the ADNI database. Beyond the inclusion criteria defined by ADNI, we applied the following additional selection criteria for the current study: (i) availability of baseline CSF sTREM2 values, (ii) no rare genetic *TREM2* variants, (iii) baseline and at least 1 follow-up [18]F-Florbetapir (AV45) amyloid PET assessment, (iv) meeting criteria as cognitively normal with a negative amyloid PET (Aβ−), or (v) showing a positive amyloid PET (i.e., Aβ+), regardless of clinical diagnosis. Clinical classification was performed by ADNI centers as cognitively normal (CN, MMSE > 24, CDR = 0, non-depressed), mild cognitively impaired (MCI; MMSE > 24, CDR = 0.5, objective memory-loss on the education adjusted Wechsler Memory Scale II, preserved activities of daily living) or AD dementia following standard diagnostic criteria (Petersen *et al*, 2010). In a subset of 54 participants, AV1451 tau PET was available. Following the recently proposed NIA-AA guidelines (Jack *et al*, 2018), we defined evidence of Alzheimer's pathologic change by biological markers (i.e., a positive amyloid PET scan), not by clinical symptoms. CSF sTREM2 measurements at baseline were assessed via ELISA by the laboratory of C. Haass at the DZNE Munich as described previously (Suarez-Calvet *et al*, 2019). The study was ethically approved by the institutional review boards of all participating ADNI centers. All ADNI participants (or their relatives) provided written informed consent. All procedures have been conducted in accordance with the declaration of Helsinki and the Department of Health and Human Services Belmont Report.

### Patient consent

Data were obtained from the multicenter study Alzheimer's disease neuroimaging initiative (AD). According to ADNI policy approved by the centers' ethics committees, all participants gave their written consent.

### Mouse model

A total of 15 $App^{NL-G-F}$ mice and 43 age- and gender-matched C57BL/6 control mice (all female) were assessed with [18]F-florbetaben amyloid PET and [18]F-GE180 TSPO-PET as described previously (Sacher *et al*, 2019). Breeding of $App^{NL-G-F}$ mice was performed at the Center for Neuropathology and Prion Research at LMU Munich. C57BL/6 mice were purchased from Charles River (Sulzberg, Germany). Animals were housed in a temperature- and humidity-controlled environment with a 12-h light–dark cycle, with free access to food and water. PET imaging sessions were performed longitudinally at 5 and 10 months of age. In brief, amyloid PET was acquired from 30 to 60 min post-injection and TSPO-PET was acquired from 60 to 90 min post-injection during 1.5% isoflurane anesthesia using a Siemens Inveon dedicated PET system. PET images of each animal were spatially co-registered to tracer specific templates and subsequently intensity-normalized using the pre-established periaqueductal

gray as a pseudo-reference region (Focke *et al*, 2019). Resulting standardized uptake value ratios (SUVRs) for both ligands were consecutively extracted from a cortical volume of interest. All experiments were performed in compliance with the National Guidelines for Animal Protection, Germany, with the approval of the regional animal committee (Regierung Oberbayern) and were overseen by a veterinarian. No randomization was used in the current study.

### Amyloid PET assessment and computation of annual amyloid PET change

AV45-PET in ADNI was acquired in a total of six 5-min time-frames 60–90 min after bolus injection of 370 Mbq radiolabeled F18-AV45 tracer. The six time-frames were subsequently co-registered and averaged to obtain a mean AV45 image. Global AV45 SUVR was assessed as the mean across selected cortical regions that was intensity-normalized to a composite reference region (including the whole cerebellum and cerebral white-matter), following a previously described protocol (Landau *et al*, 2015). The composite reference region was chosen since it has been previously shown to yield to most stable estimates of longitudinal AV45 changes (Landau *et al*, 2015). For each participant, we then determined the annual changes between subsequently assessed global AV45 SUVR values (i.e., absolute SUVR difference divided by the time difference in years). Thus, for participants with more than two AV45 scans, more than one AV45 change rate was determined (e.g., from baseline to follow-up 1 and from follow-up 2 to follow-up 3). All AV45 change rates were subsequently transformed to z-scores with a mean of 0 and a standard deviation of 1.

### AV1451 tau PET assessment

Tau PET was assessed 75 min after bolus injection of F18-radiolabeled AV1451 tracer in 6 × 5 min blocks. Recorded images were co-registered and averaged across blocks and intensity-normalized to the inferior cerebellar gray (Maass *et al*, 2017). SUVR scores for Braak-stage-specific ROIs were derived by the ADNI core and downloaded from the ADNI database. Detailed protocols can be found on the ADNI homepage and in previous publications (Scholl *et al*, 2016). Note that we excluded Braak-stage 2 (i.e., hippocampus) due to known off-target binding of the AV1451 tracer in this region (Lemoine *et al*, 2018). The cross-sectionally collected AV1451 tau PET SUVR data (i.e., intensity-normalized to the inferior cerebellar gray) were acquired maximum 3 years (mean = 1.6 years) after the sTREM2 assessments and the first available AV45-PET scan.

### Statistical analysis

Sample characteristic was compared between study groups using chi-squared tests for categorical measures and ANOVAS for continuous measures.

To assess whether the baseline AV45 PET shows an u-shaped association with the rate of AV45 PET accumulation, we computed linear mixed models with either global AV45 PET or global AV45 $PET^2$ as a predictor of subsequent AV45 PET change, controlling for age, gender, education, diagnosis, CSF p-tau$_{181}$, ApoE4 genotype (ε4 carrier vs. non-carrier), time between AV45 PET visits, and random intercept. Note that linear AV45 effects were also included as a fixed

effect when testing quadratic (i.e., AV45 $PET^2$) effects. Model fit between the linear (i.e., AV45 PET) and the quadratic (i.e., AV45 $PET^2$) predictors was compared using likelihood ratio tests. Next, we tested our main hypothesis that higher sTREM2 levels are associated with slower amyloid accumulation. To this end, we used linear mixed models to test whether global (i) AV45 $PET^2$ and sTREM2 had independent main effects on subsequent AV45 changes, or (ii) whether there was a global AV45 $PET^2 \times$ sTREM2 interaction (i.e., quadratic) on subsequent AV45 PET changes, controlling for the linear interaction term (i.e., AV45 $\times$ sTREM2) as well as age, gender, education, diagnosis, CSF p-tau$_{181}$, ApoE4, time between AV45 PET visits, and random intercept. In order to test at which phase of amyloid accumulation, any potential effect of CSF sTREM2 is highest, we performed sliding window analyses on the association between sTREM2 and future amyloid accumulation at different levels of baseline AV45 levels. Specifically, we rank-ordered the subject-wise global amyloid PET-values from low to high and group subjects into batches of 100 participants at a step size of $n = 10$, yielding 62 batches. Within each window (i.e., batch), we then split the participants between high and low sTREM2 levels (i.e., via median split, $n = 50$) and estimated the effect size Cohen's $d$ for the difference in the rate of global amyloid PET change between high and low CSF sTREM2 groups.

To validate the effect of a neuroimmune response on amyloid accumulation in a preclinical AD model, we tested whether higher microglial activation (i.e., GE180 PET) was associated with slower amyloid accumulation (i.e., Florbetaben PET) in the $App^{NL-G-F}$ mouse model of AD. Here, we first assessed whether $App^{NL-G-F}$ AD mice and C57BL/6 controls had higher Florbetaben and GE180 PET uptake at 5 months of age using two-sample $t$-tests. Second, we tested in the $App^{NL-G-F}$ AD mice, whether higher baseline GE180 PET at 5 months of age was associated with slower rate of Florbetaben PET increase from 5 to 10 months of age, using linear regression controlling for baseline Florbetaben PET.

In exploratory analyses, we tested whether higher sTREM2 levels were also associated with lower tau levels. To this end, we included AV1451 tau PET that was available for an ADNI subsample ($n = 54$) within 3 years of the sTREM2 assessments. Using ANCOVAs, we tested whether higher sTREM2 levels (i.e., defined via median split) were associated with lower tau PET SUVR in Braak-stage-specific ROIs, controlling for age, gender, education, diagnosis, ApoE4 as well as global AV45 amyloid PET and time between the sTREM2 and tau PET assessments. All analyses were performed in R statistical software (Version 3.6.1).

## Data availability

The dataset supporting the conclusions of this article is available at the ADNI homepage (https://ida.loni.usc.edu).

Expanded View for this article is available online.

## Acknowledgements
This work was supported by LMUexcellent (to M.E), and the Deutsche Forschungsgemeinschaft (DFG) within the framework of the Munich Cluster for Systems Neurology (EXC 1010 SyNergy), a DFG funded Koselleck Project (HA1737/16-1 to C.H.) and the FTD Biomarker Award. Data collection and sharing for this project was funded by the Alzheimer's Disease Neuroimaging Initiative (ADNI) (National Institutes of Health Grant U01 AG024904) and DOD ADNI (Department of Defense award number W81XWH-12-2-0012). ADNI is funded by the National Institute on Aging, the National Institute of Biomedical Imaging and Bioengineering, and through generous contributions from the following: AbbVie, Alzheimer's Association; Alzheimer's Drug Discovery Foundation; Araclon Biotech; BioClinica, Inc.; Biogen; Bristol-Myers Squibb Company; CereSpir, Inc.; Cogstate; Eisai Inc.; Elan Pharmaceuticals, Inc.; Eli Lilly and Company; EuroImmun; F. Hoffmann-La Roche Ltd and its affiliated company Genentech, Inc.; Fujirebio; GE Healthcare; IXICO Ltd.; Janssen Alzheimer Immunotherapy Research & Development, LLC.; Johnson & Johnson Pharmaceutical Research & Development LLC.; Lumosity; Lundbeck; Merck & Co., Inc.; Meso Scale Diagnostics, LLC.; NeuroRx Research; Neurotrack Technologies; Novartis Pharmaceuticals Corporation; Pfizer Inc.; Piramal Imaging; Servier; Takeda Pharmaceutical Company; and Transition Therapeutics. The Canadian Institutes of Health Research is providing funds to support ADNI clinical sites in Canada. Private sector contributions are facilitated by the Foundation for the National Institutes of Health (www.fnih.org). The grantee organization is the Northern California Institute for Research and Education, and the study is coordinated by the Alzheimer's Therapeutic Research Institute at the University of Southern California.

### The paper explained

#### Problem
Microglia activation forms the brain's innate immune response to cerebral pathologies including fibrillar beta-amyloid (Aβ), i.e., a primary pathology in Alzheimer's disease. TREM2 is a receptor protein expressed in the brain by microglia, where signaling through TREM2 triggers the phagocytosis of pathogens such as Aβ. Since microglia activation such as that triggered by TREM2 activation constitutes a potential drug target, understanding the role of increased TREM2 and microglia activation in AD progression is pivotal. Here, we tested the role of microglia activation in a two-pronged approach. First, we assessed cerebrospinal fluid (CSF) biomarker levels of the soluble fraction of TREM2 (sTREM2) as a predictor of longitudinal increase in Aβ as assessed by amyloid PET imaging over 2 years on average in non-demented and demented participants. Second, in order to confirm a link between higher microglia activation and changes in Aβ accumulation, we tested microglia activation (assessed by microglia PET imaging) as a predictor of rates of amyloid PET increase in a mouse model of genetically caused Aβ.

#### Results
We found that higher baseline CSF sTREM2 levels were associated with a slower rate of amyloid PET accumulation, in particular during phases of intermediate baseline Aβ levels in elderly subjects. In the transgenic mouse model of Aβ, we found that higher microglia PET at 5 months of age, i.e., when Aβ accumulation starts to emerge globally in the brain, to be associated with reduced rates of amyloid PET during a phase until 10 months of age, i.e., a period when Aβ is strongly increasing.

#### Impact
We conclude that higher microglia activation and in particular TREM2 during a period of intermediate baseline levels of Aβ are associated with reduced rates of Aβ accumulation. These results suggest that enhancing TREM2-related microglia activation may exert protective effects on Aβ.

ADNI data are disseminated by the Laboratory for Neuro Imaging at the University of Southern California. YD is supported by an NIMH institutional training grant (T32MH014877).

## Author contributions

MB, MS-C, GB, CS, TB, EM-R, GK, YD, JH, MD, and LP performed the experiments. MS-C, NF, YD, ME, and CH analyzed and interpreted the data. YD, LP, and CC were involved in extracting the genetic data. LS, and JQT. MB contributed with patient samples and/or data. MS-C, MB, ME, CH, and NF designed the study and wrote the manuscript. All authors critically reviewed and approved the final manuscript.

## Conflicts of interest

C.H. collaborates with DENALI Therapeutics. G.K. is an employee of ISAR Bioscience. The remaining authors declare that they have no conflict of interest (financial or otherwise).

## For more information

(i) Ewers Lab: https://www.isd-research.de/ewers-lab
(ii) Haass Lab: https://www.biochemie.abi.med.uni-muenchen.de/haass1/index.html
(iii) ADNI: http://adni.loni.usc.edu

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
