## [Review Process File · EMBO Molecular Medicine]

Higher CSF sTREM2 and microglia activation associate with slower rate of beta-amyloid accumulation

Michael Ewers, Gloria Biechele, Marc Suárez-Calvet, Christian Sacher, Tanja Blume, Estrella Morenas-Rodriguez, Yuetiva Deming, Laura Piccio, Carlos Cruchaga, Gernot Kleinberger, Leslie Shaw, Jochen Herms, Martin Dichgans, Matthias Brendel, Christian Haass, and Nicolai Franzmeier
DOI: 10.15252/emmm.202012308

Corresponding authors: Michael Ewers (Michael.Ewers@med.uni-muenchen.de) , Christian Haass (christian.haass@mail03.med.uni-muenchen.de)

Review Timeline:

Submission Date:	9th Mar 20
Editorial Decision:	28th Apr 20
Revision Received:	28th May 20
Editorial Decision:	9th Jun 20
Revision Received:	3rd Jul 20
Accepted:	10th Jul 20

Editor: Jingyi Hou

Transaction Report:

28th Apr 2020

Dear Prof. Ewers,

Thank you for the submission of your manuscript to EMBO Molecular Medicine. We have now received feedback from two out of three referees whom we asked to evaluate your manuscript. Given that both reviewers provide similar recommendations, we prefer to make a decision now in order to avoid further delay in the process. As you will see from the reports below, the referees acknowledge the potential interest of the study. However, they also raise substantial concerns about your work, which should be convincingly addressed in a major revision of the present manuscript.

During our pre-decision cross-commenting process (in which the reviewers are given the chance to make additional comments, including on each other's reports), referee #3 added "With regard to the mice part of this study, this suggestion is only applicable if the data is already available, I do not think this is a requirement for the present study."

I think that the referees' recommendations are rather clear and there is no need to reiterate their comments. Importantly, clarification and refinement of data analysis would be necessary. We would encourage you to add on exploratory analysis with Tau as suggested by referee #1 to further enhance the impact of the study, but this is not mandatory for publication.

We would welcome the submission of a revised version within three months for further consideration. Please note that EMBO Molecular Medicine strongly supports a single round of revision and that, as acceptance or rejection of the manuscript will depend on another round of review, your responses should be as complete as possible.

We are aware that many laboratories cannot function at full efficiency during the current COVID-19/SARS-CoV-2 pandemic and have therefore extended our "scooping protection policy" to cover the period required for a full revision to address the experimental issues. Please let me know should you need additional time, and also if you see a paper with related content published elsewhere.

I look forward to receiving your revised manuscript.

Yours sincerely,
Jingyi Hou

Jingyi Hou
Editor
EMBO Molecular Medicine

*** Instructions to submit your revised manuscript ***

** PLEASE NOTE ** As part of the EMBO Publications transparent editorial process initiative (see our Editorial at <https://www.embopress.org/doi/pdf/10.1002/emmm.201000094>), EMBO Molecular Medicine will publish online a Review Process File to accompany accepted manuscripts.

To submit your manuscript, please follow this link:

Link Not Available

- 1) a .doc formatted version of the manuscript text (including Figure legends and tables). Please make sure that the changes are highlighted to be clearly visible to referees and editors alike.
- 2) separate figure files*
- 3) supplemental information as Expanded View and/or Appendix. Please carefully check the authors guidelines for formatting Expanded view and Appendix figures and tables at <https://www.embopress.org/page/journal/17574684/authorguide#expandedview>
- 4) a letter INCLUDING the reviewers' reports and your detailed responses to their comments (as Word file)

Also, and to save some time should your paper be accepted, please read below for additional information regarding some features of our research articles:

- 5) The paper explained: EMBO Molecular Medicine articles are accompanied by a summary of the articles to emphasize the major findings in the paper and their medical implications for the non-specialist reader. Please provide a draft summary of your article highlighting
 - the medical issue you are addressing,

- the results obtained and
- their clinical impact.

6) For more information: There is space at the end of each article to list relevant web links for further consultation by our readers. Could you identify some relevant ones and provide such information as well? Some examples are patient associations, relevant databases, OMIM/proteins/genes links, author's websites, etc...

7) Author contributions: the contribution of every author must be detailed in a separate section (before the acknowledgments).

8) EMBO Molecular Medicine now requires a complete author checklist (<https://www.embopress.org/page/journal/17574684/authorguide>) to be submitted with all revised manuscripts. Please use the checklist as a guideline for the sort of information we need WITHIN the manuscript as well as in the checklist. This is particularly important for animal reporting, antibody dilutions (missing) and exact p-values and n that should be indicated instead of a range.

9) Every published paper now includes a 'Synopsis' to further enhance discoverability. Synopses are displayed on the journal webpage and are freely accessible to all readers. They include a short stand first (maximum of 300 characters, including space) as well as 2-5 one sentence bullet points that summarise the paper. Please write the bullet points to summarise the key NEW findings. They should be designed to be complementary to the abstract - i.e. not repeat the same text. We encourage inclusion of key acronyms and quantitative information (maximum of 30 words / bullet point). Please use the passive voice. Please attach these in a separate file or send them by email, we will incorporate them accordingly.

You are also welcome to suggest a striking image or visual abstract to illustrate your article. If you do please provide a jpeg file 550 px-wide x 400-px high.

10) A Conflict of Interest statement should be provided in the main text

11) Please note that we now mandate that all corresponding authors list an ORCID digital identifier. This takes <90 seconds to complete. We encourage all authors to supply an ORCID identifier, which will be linked to their name for unambiguous name identification.

Currently, our records indicate that there is no ORCID associated with your account.

Please click the link below to provide an ORCID:

Link Not Available

12) The system will prompt you to fill in your funding and payment information. This will allow Wiley to send you a quote for the article processing charge (APC) in case of acceptance. This quote takes into account any reduction or fee waivers that you may be eligible for. Authors do not need to pay any fees before their manuscript is accepted and transferred to our publisher.

Photos 400-800 DPI

*Additional important information regarding figures and illustrations can be found at <http://bit.ly/EMBOPressFigurePreparationGuideline>

***** Reviewer's comments *****

Referee #1 (Remarks for Author):

Although increased CSF sTREM2 previously have been associated with reduced disease activity in AD, this is the first study to examine it's role on longitudinal Ab accumulation. The finding that increased sTREM2 is statistically associated with reduced Ab accumulation is interesting but I feel that the finding would be easier to understand if further subanalyses are included.

1. My main concern relates to whether there actually is any causal relationship between TREM2 and Ab accumulation. I note that the authors make no causal claim but I still believe that the paper would benefit from trying to tackle this problem (a mere statistical association is of less interest). Can the authors e.g. differ between hypothesis 1: Does sTREM2 indeed affect Ab accumulation? Or hypothesis 2: Is it just a disease stage marker that is high in stages where the accumulation rate is lower (but does not really affect ab accumulation directly)? One way of understanding this better would be to run analysis separately in CN Ab+, MCI Ab+ and AD. If the association between sTREM2 and Ab accumulation rate exists in all phases of the disease this would favour hypothesis 1. As it is now, the authors only include a sub analysis in Ab+ MCI and AD (i.e. late disease stages in terms of amyloidosis), but not CN Ab+. As for clinical trials and future medical decision regarding whom to treat with anti-Ab therapies, the predictive role of sTREM2 is without doubt most interesting in CN Ab+ individuals (predicting Ab accumulation rate has less relevance in symptomatic individuals).

2. The authors calculate two AV45 change rates in those with more than one scan? Does that mean that they, when predicting the rate between scan 2 and 3, use scan 2 as "baseline" scan/predictor? If not, would it not make more sense to use a coefficient based on all available scans (which should produce a more stable measure of the accumulation rate)? Or would this affect the ability test the U-shape hypothesis?

3. As a shortcoming, the authors mention that they only study the role of sTREM2 on Ab but not tau. Since the paper contains quite few analyses, I suggest that they include an exploratory analysis testing the effect of sTREM2 on tau using tau PET (n=54) and/or CSF P-tau (n=200+). I know that the CSF P-tau data in ADNI isn't the best but it is worth a shot and would definitely increase the impact of the study.

4. It would be interesting if the authors briefly could mention the results in relation to other inflammatory CSF biomarkers such as YKL-40.

Referee #3 (Remarks for Author):

This study investigated the association of sTREM2 levels in CSF on the rate of amyloid aggregation on PET in human beings, and the relationship of microglial PET signal and amyloid PET in AD mouse models (APP-NL-G-F mice). Although sTREM2 is not determined in the mice, it is interesting to put these analyses next to patient data. Overall this study is well designed and written, and provides new insights into the role of microglia activation in inter-individual differences in amyloid aggregation. I have the following questions/suggestions:

Statistical models:

It is posited that amyloid aggregation shows an inverted U curve, and it is hypothesised that the protective effect of sTREM2 is specific for the initial increase of this U curve (in other words will slow aggregation): in this part of the curve higher sTREM2 CSF levels may reflect activated microglia that are clearing the plaques. I have some questions about how these relationships modelling part, which was not so clear in the methods section:

-Linear mixed models are used to assess the effects of baseline amyloid PET levels on subsequent annual change on amyloid PET, but there is no description how time is included in these mixed models. Did the authors test interactions of baseline PET values x time on the repeated PET? How was baseline PET² next included, also as interaction term with time? And similarly, the last equation baseline PET² x time x sTREM2?

-For the quadratic relationships the description in the methods seems to suggest that the model only included either baseline PET or baseline PET², but for modelling a quadratic relationship, both ¹ and ² need to be included.

-Model fit of linear vs quadratic was performed by comparing the AIC, however, it is more common to perform a likelihood ratio test to further quantify whether models are different/better fitting the data.

-But, possibly, linear mixed models were first used to determine subject specific slopes, which were used as input to model the baseline PET relationship with, as well as with CSF sTREM levels in linear regression models? (this is what seems to be suggested from p8 and the figures, but the exact statistics how are not so clearly described....). P8 also suggests that multiple change rates can be estimated for individuals (which would explain why a random effect was included in the subsequent models), but, amyloid PET is noisy data and why not make use of having more time points to make more accurate estimates of change rates? Which brings me to the next question:

-The models describe that the linear mixed models included random intercepts only, why not also include random slopes: it is the differences amongst individuals in rates of aggregation that are the main focus of this study, why not allow slopes to vary amongst individuals?

-Have the authors considered to test spline models to study whether the associations of sTREM2 levels with rate of aggregation depend on whether individuals are before or after the peak of aggregation? For example general additive models may help with this.

-Figure 1 shows the relationships between baseline levels and annual change on amyloid PET for the high/low sTREM groups. It is not immediately clear why the high levels are protective in initial aggregating phase, as for this group the relationship between baseline and annual change on PET seems approximately flat (i.e., same rate regardless of baseline amyloid values), and the confidence intervals are mostly overlapping between the sTREM groups. Would it be possible to define the peak of the amyloid aggregation vs baseline curve, and provide separate scatterplots for each side of the peak for annual AV45 change against sTREM2 levels?

-Was Cohen's d calculated by transforming the beta coefficient? For the present analyses, wouldn't variance explained be a more intuitive measure, since relationships between continuous measures are tested, rather than group comparisons?

-What software was used to perform the statistical analyses?

Other questions:

-the APP NLGF models display microgliosis and astrogliosis, as is also reflected on PET in this study. However, this model does not show neurofibrillary tangles or neuronal loss according to Saito et al., (2014), making it difficult to interpret results as potentially protective effects. How can the argument of protective effects of activated microglia in this mouse model further be supported, other than changes on amyloid PET?

-Figure 2 shows overlap between the wild-type and APP-NLGF mice both for the microglial activation and the rate of change in amyloid PET signals. How can this overlap be interpreted, could this reflect technical issues? Could the wildtype data be added to the scatterplot to show whether these also show a similar relationship as the APP-NLGF mice.

-Has CSF been collected in these mice or in this model, would it be possible to measure sTREM2 for them as well, to provide more support for the comparison between humans & model (this is not a suggestion to perform if unavailable, in that case perhaps an idea for the future)?

-Perhaps the authors could provide more discussion on potential causes of inter-mice differences in microglial activation signal, given that they are genetically identical. One suggestion is given, and discarded (i.e., influence of enhanced environment - but I am assuming these mice were reared in similar environments, which were not necessarily enhanced?)

Explanation of Changes

Referee #1 (Remarks for Author):

Although increased CSF sTREM2 previously have been associated with reduced disease activity in AD, this is the first study to examine it's role on longitudinal Ab accumulation. The finding that increased sTREM2 is statistically associated with reduced Ab accumulation is interesting but I feel that the finding would be easier to understand if further subanalyses are included.

1. Comment:

My main concern relates to whether there actually is any causal relationship between TREM2 and Ab accumulation. I note that the authors make no causal claim, but I still believe that the paper would benefit from trying to tackle this problem (a mere statistical association is of less interest). Can the authors e.g. differ between hypothesis 1: Does sTREM2 indeed affect Ab accumulation? Or hypothesis 2: Is it just a disease stage marker that is high in stages where the accumulation rate is lower (but does not really affect ab accumulation directly)? One way of understanding this better would be to run analysis separately in CN Ab+, MCI Ab+ and AD. If the association between sTREM2 and Ab accumulation rate exists in all phases of the disease this would favour hypothesis 1. As it is now, the authors only include a sub analysis in Ab+ MCI and AD (i.e. late disease stages in terms of amyloidosis), but not CN Ab+. As for clinical trials and future medical decision regarding whom to treat with anti-Ab therapies, the predictive role of sTREM2 is without doubt most interesting in CN Ab+ individuals (predicting Ab accumulation rate has less relevance in symptomatic individuals).

Response: The reviewer raises an important point, namely whether 1) higher sTREM2 levels – as a proxy of microglial activation - are causally linked to lower amyloid accumulation, or whether 2) elevations in sTREM2 simply show a temporal contiguity with phases of low amyloid accumulation

We agree that the question of a causal relationship merits further consideration, even though any conclusions must be limited as the current study is correlational in nature.

To address the question, the reviewer suggests testing our interaction models stratified by diagnostic groups. The underlying assumption is that clinical staging is a good proxy for differences in the rate of amyloid accumulation. However, when testing group differences in the rate of global amyloid PET, we did not find significant ($p > 0.05$) group-differences in AV45 change rates across Ab+ groups (i.e. CN-A β + vs. MCI A β + vs. AD Dementia) in ANCOVAs controlling for age, gender, education and ApoE4. This suggests, that the rate of A β accumulation is mainly a function of baseline levels of A β (as illustrated in Figure 1A in the original manuscript), rather than clinical stage. Therefore, in order to address the reviewer's question, we chose to test the effect of CSF sTREM2 on the rate of amyloid at different levels of baseline AV45 PET rather than different clinical stages. To this end, we used a sliding time window approach, where within each "window" of AV45 PET values, we estimated the size of the effect of CSF sTREM2 on the rate of change of AV45 PET. Thus, if there is a mere coincidence of high CSF sTREM2 vs low AV45 PET accumulation rates, the effect size should be largest in those windows of baseline AV45 PET, where the rate of AV45 PET accumulation is low. Briefly, we rank-ordered the subject-level global amyloid PET-values from low to high and grouped subjects into batches of 100 participants at a step size of $n = 10$, yielding 62 batches. Within each window (i.e. batch), we then split the participants

between high and low sTREM2 levels (i.e. via median split, $n = 50$) and estimated the effect size Cohen's d for the difference in the rate of global amyloid PET change between high and low CSF sTREM2 groups.

The results showed that the strongest effect size of sTREM2 was observed at the baseline amyloid PET level of 1.03, which is close to that of the peak rate of amyloid accumulation (i.e. at baseline amyloid PET = 0.95 corresponding to dashed line in the figure below). This result speaks against the hypothesis that the observed protective effects of CSF sTREM2 merely result from a coincidental occurrence of lower rates of amyloid accumulation and high levels of CSF sTREM2. Rather, the higher sTREM2 are associated with reduced rates of amyloid PET accumulation at a disease stage when $A\beta$ accumulation thrives. This result does not show a causal relationship but discounts the hypothesis that the current findings result from high levels of CSF sTREM2 during when $A\beta$ accumulation during the disease course.

We have added this analysis to the statistics (p.10) and results part of the manuscript (p.14 & supplementary figure 1) and Discussion (p. 22).

2. Comment:

The authors calculate two AV45 change rates in those with more than one scan? Does that mean that they, when predicting the rate between scan 2 and 3, use scan 2 as "baseline" scan/predictor? If not, would it not make more sense to use a coefficient based on all available scans (which should produce a more stable measure of the accumulation rate)? Or would this affect the ability to test the U-shape hypothesis?

Response: The reviewer is correct that we calculated more than one AV45 change rate in individuals with more than two scans. Specifically, the change rate between scan 1 and scan 2 uses scan 1 as a baseline, whereas the change rate between scan 2 and 3 uses scan 2 as a baseline. We specifically chose this approach of calculating multiple change rates per patient, in order to be able to assess the rate of amyloid accumulation in the near future at a given level of amyloid deposition, which is best-suited to test the U-shape hypothesis. We agree,

however, that computing annual AV45 change rates across all available scans is an alternative way to test the current hypothesis. Thus, we altered the analysis, this time using linear regression testing the $AV45^2 \times sTREM2 + AV45 \times sTREM2$ effects (plus main effects of each term) on the annual AV45 change rate computed across all available scans. Like the linear mixed effects models described in the manuscript, this linear regression was controlled for age, gender, education, diagnosis, ApoE4 and CSF p-tau levels. For the quadratic interaction $AV45^2 \times sTREM2$, we find consistent effects with our main results ($T=-2.005$, $p=0.046$). However, the effects are probably less strong due to the reduced degrees of freedom and suboptimal modeling of the U-shaped curve of amyloid accumulation. Still, these results confirm the analyses presented in the manuscript, suggesting that higher sTREM2 levels are associated with slower rates of amyloid accumulation. Since we believe that the initial analyses computing AV45 change rates between two subsequent scans is best suited to test the inverse-U-shape hypothesis, we kept this analysis in the manuscript.

3. Comment:

As a shortcoming, the authors mention that they only study the role of sTREM2 on Ab but not tau. Since the paper contains quite few analyses, I suggest that they include an exploratory analysis testing the effect of sTREM2 on tau using tau PET (n=54) and/or CSF P-tau (n=200+). I know that the CSF P-tau data in ADNI isn't the best, but it is worth a shot and would definitely increase the impact of the study.

Response: We appreciate the reviewer's suggestion to assess the association between sTREM2 and tau levels. To address the reviewers concern, we conducted exploratory analyses using cross-sectional AV1451 tau-PET SUVR data (i.e. intensity normalized to the inferior cerebellar grey) that was acquired maximum 3 years (mean=1.6 years) after the sTREM2 assessments and the first available AV45-PET scan. Given that tau PET is a more direct measure of neurofibrillary tangles, whereas CSF p-tau only moderately correlates with tau PET and may represent increased levels of soluble tau production, we limited our additional analysis to the tau PET measure (Mattsson-Carlsson et al., 2020).

The AV1451 tau-PET data were available for a total of 54 subjects including cognitively normal Ab-, CN A β + and MCI A β + (see updated table 1, p. 42). Freesurfer-based AV1451-PET SUVR scores were downloaded from the ADNI database for each of the three composite ROI including Braak stage 1, Braak stage 3+4 and Braak stage 5+6 (Scholl et al., 2016). Note that Braak stage 2 (i.e. Hippocampus) was excluded from the analyses due to known off-target binding of the AV1451 tracer in this region (Lemoine, Leuzy, Chiotis, Rodriguez-Vieitez, & Nordberg, 2018). Using ANCOVAs, we tested whether higher CSF sTREM2 levels at baseline (i.e. defined via median split) were associated with lower AV1451-PET, controlling for baseline AV45 amyloid PET as well as age, gender, education, ApoE4 status, diagnosis and time between the CSF sTREM2 assessments and the AV1451-PET scan. Here, we found that higher CSF sTREM2 levels were associated with lower AV1451-PET in Braak 1 ($F=9.527$, $p=0.005$) as well as Braak 3+4 ($F=5.253$, $p=0.032$). For Braak 5+6, no significant differences in AV1451-PET were observed between subjects with high vs. low CSF sTREM2 levels at baseline ($F=2.810$, $p=0.108$). After applying Bonferroni correction for 3 tests ($\alpha=0.017$), only AV1451-PET levels in Braak stage 1 ROIs (i.e. entorhinal cortex) remained significantly different between subjects with high vs. low sTREM2. Together, this suggests that higher sTREM2 levels are associated with lower PET-assessed tau levels in early Braak stage regions, independent of A β levels. Our results are consistent with previous reports of a recent study in a transgenic mouse model of A β , showing that genetically induced TREM2 deficiency was associated with higher pathologic tau seeding in neurotic plaques (Leyns et al., 2019). We have added these exploratory analyses to the methods (p.9), statistics

(p.11) and results section (p.14, Figure 5), and briefly introduce the analysis (p. 6) and discuss the results (p.19).

4. Comment:

It would be interesting if the authors briefly could mention the results in relation to other inflammatory CSF biomarkers such as YKL-40.

Response: Measures of CSF YKL-40 were not available in ADNI. YKL-40 is primarily expressed by astrocytes and microglia in the brain, where CSF levels of YKL-40 have been previously reported to be increased in Alzheimer's disease, peaking at the prodromal stage (Gispert et al., 2016), similarly to what has been described for CSF sTREM2 (Suarez-Calvet et al., 2016). It remains to be investigated whether YKL-40 regulates the rate of amyloid pathology, which we could not address in the current study.

A number of other inflammatory markers can be detected in the CSF such as cytokines. However, we believe that any such exploratory endeavor would be beyond the scope of the current hypothesis-driven study on CSF sTREM2.

We now briefly mention YKL-40 as a potential additional marker in the Discussion (p. 23)

Referee #3 (Remarks for Author):

This study investigated the association of sTREM2 levels in CSF on the rate of amyloid aggregation on PET in human beings, and the relationship of microglial PET signal and amyloid PET in AD mouse models (APP-NL-G-F mice). Although sTREM2 is not determined in the mice, it is interesting to put these analyses next to patient data. Overall this study is well designed and written, and provides new insights into the role of microglia activation in inter-individual differences in amyloid aggregation. I have the following questions/suggestions:

1. Comment:

It is posited that amyloid aggregation shows an inverted U curve, and it is hypothesised that the protective effect of sTREM2 is specific for the initial increase of this U curve (in other words will slow aggregation): in this part of the curve higher sTREM2 CSF levels may reflect activated microglia that are clearing the plaques. I have some questions about how these relationships modelling part, which was not so clear in the methods section:

Linear mixed models are used to assess the effects of baseline amyloid PET levels on subsequent annual change on amyloid PET, but there is no description how time is included in these mixed models. Did the authors test interactions of baseline PET values x time on the repeated PET? How was baseline PET² next included, also as interaction term with time? And similarly, the last equation baseline PET² x time x sTREM2?

Response: We are happy to clarify the regression models. As the reviewer rightly alludes to in a later comment, we first estimated subject specific slopes, before entering them as dependent variables in the model. In order to assess the annual rate of change in amyloid PET (i.e. slope) per subject, we computed the difference in global amyloid PET uptake between any two consecutive time points for each subject. That is, we first computed AV45 PET SUVR change rates defined as scan 2 scan minus scan 1 (i.e. “baseline”) divided by the time difference between scans in years **or** scan 3 minus scan 2 (i.e. “baseline”) divided by the time difference between scans in years etc. Accordingly, we computed one change rate for individuals with 2 AV45 PET scans, two change rates for individuals with 3 AV45 PET scans, and 3 change rates for individuals with 4 AV45 PET scans (A detailed description of how we computed annual change rates between two subsequent AV45 scans can be found in the paragraph “*Amyloid-PET assessment and computation of annual amyloid-PET change*” on p.8 of the methods section).

In order to test the first hypothesis, i.e. that the AV45 PET change rate shows an inverse u-shaped curve as a function of baseline AV45 PET, we tested the effects of baseline AV45 PET² + baseline AV45 on subsequent AV45 change rates.

In order to test our main hypothesis that sTREM2 attenuates the u-shaped association between baseline AV45 PET and subsequent AV45 change, we included in addition, the interaction terms sTREM2 x baseline AV45² + sTREM2 x baseline AV45. In all analyses, we included random intercepts for subjects to account for differences in the number of available timepoints per subject. Note that random slopes (e.g. time as a random factor) were not included, since slopes were pre-estimated for each subject. To control for any differences in the time interval between two subsequent AV45 PET scans, all analyses described above further included the duration of the time interval between AV45 scans in years as a fixed covariate.

As an alternative approach (in response to reviewer 1, comment 2), we now estimated the annual rate of change in AV45 PET across all available AV45 visits (i.e. one change rate per subject, see also response to comment 2 by reviewer 1 above). Using these data, we tested again the effects of baseline AV45 PET² x sTREM2 + baseline AV45 x sTREM2 on annual

AV45 change rates using linear regression and the same covariates as in the linear mixed models. A random effect of the number of slopes was no longer included since the number of available change rates was fixed to N=1 per subject. In this linear regression analysis, we find a consistent AV45² x sTREM2 interaction on annual change rates in AV45 (p=0.047).

2. Comment:

For the quadratic relationships the description in the methods seems to suggest that the model only included either baseline PET or baseline PET², but for modelling a quadratic relationship, both ¹ and ² need to be included.

Response: Thank you for bringing up that point. We like to clarify that all models in the originally submitted version of the manuscript that tested a quadratic association also included the linear terms. We have added this missing information in the Methods where pertinent (p.10).

3. Comment:

Model fit of linear vs quadratic was performed by comparing the AIC, however, it is more common to perform a likelihood ratio test to further quantify whether models are different/better fitting the data.

Response: We thank the reviewer for this comment. We have now applied likelihood ratio tests to compare linear vs. quadratic models. Here, all results remained consistent with our previous analyses using AIC, showing that quadratic effects showed a better model fit than linear effects. We have included the likelihood ratio tests in the methods (p.10) and results sections (p.12-13) of the manuscript.

4. Comment:

But, possibly, linear mixed models were first used to determine subject specific slopes, which were used as input to model the baseline PET relationship with, as well as with CSF sTREM2 levels in linear regression models? (this is what seems to be suggested from p8 and the figures, but the exact statistics how are not so clearly described....). P8 also suggests that multiple change rates can be estimated for individuals (which would explain why a random effect was included in the subsequent models), but, amyloid PET is noisy data and why not make use of having more time points to make more accurate estimates of change rates? Which brings me to the next question:

Response: Please see our earlier response to the first comment as well as our response to comment 2 by reviewer 1, who raised the same point.

5. Comment:

The models describe that the linear mixed models included random intercepts only, why not also include random slopes: it is the differences amongst individuals in rates of aggregation that are the main focus of this study, why not allow slopes to vary amongst individuals?

Response: The reviewer is correct that the models presented in the manuscript included random intercepts but no random slopes for each subject. The reason for not including random slopes is based on the design of our statistical models: Specifically, we determined the slope of longitudinal amyloid change a priori (i.e. the annual change in AV45 SUVR between any two subsequent AV45 visits). The resulting annual AV45 change rates (i.e.

slopes) were then entered into the linear mixed models as a dependent variable. Thus, we did not include random slopes in the reported models.

However, we ran additional models, where we did not pre-define subject specific slopes in amyloid accumulation. Rather, we tested the three-way interaction of baseline amyloid x baseline sTREM2 x time on amyloid accumulation, controlling for age, gender, education, diagnosis, CSF p-tau, ApoE4 and random slope and intercept. Here, we found a significant three-way interaction effect ($T=-2.056$, $p=0.041$) that is in line with our main analyses presented in the manuscript (i.e. using pre-specified slopes of amyloid accumulation as a dependent variable). Since the two-way interaction analysis using pre-defined slopes per subject is easier to illustrate, we preferred, however, to keep this analysis in the manuscript.

6. Comment:

Have the authors considered to test spline models to study whether the associations of sTREM2 levels with rate of aggregation depend on whether individuals are before or after the peak of aggregation? For example general additive models may help with this.

Response: The reviewer raises an important point, namely to identify phases of amyloid-deposition in which sTREM2 effects on future amyloid accumulation are highest. The reviewer suggests spline models or general additive models to determine local effects of sTREM2 levels on subsequent amyloid accumulation. However, we believe that spline models may require a larger sample size in order to reliably estimate local slopes between sTREM2 levels and subsequent amyloid accumulation.

Rather than using general additive models or spline models, we thus decided to perform a sliding window analysis on the association between sTREM2 effects and the rate of amyloid accumulation across subjects from low to high baseline amyloid levels (see also response to comment 1 by reviewer 1). The aim was to assess the effect size of CSF sTREM2 on amyloid PET change as a function of baseline amyloid PET (i.e. disease severity marker). To this end, all participants were rank ordered based on the baseline amyloid PET value and grouped into batches with a window size = 100 participants. The windows were moved across the AV45 PET values at a step size of 10, rendering 62 windows. Within each window of 100 subjects, we then split subjects between high and low sTREM2 levels (i.e. via median split) and determined the difference in the rate of AV45 accumulation between high and low sTREM2 groups via the effect size Cohens d . We then plotted the resulting effect sizes against mean baseline AV45 PET within each window (figure below). We found that beneficial effects of sTREM2 on future amyloid accumulation (i.e. positive Cohen's d values) were highest in windows clustered around the peak of annual amyloid changes (i.e. at baseline AV45 PET = 0.95, indicated as the vertical line in attached figure & supplementary figure 1).

We have added this analysis to the statistics (p.10) and results part of the manuscript (p.14 & supplementary figure 1) and Discussion (p. 22).

7. Comment:

Figure 1 shows the relationships between baseline levels and annual change on amyloid PET for the high/low sTREM groups. It is not immediately clear why the high levels are protective in initial aggregating phase, as for this group the relationship between baseline and annual change on PET seems approximately flat (i.e., same rate regardless of baseline amyloid values), and the confidence intervals are mostly overlapping between the sTREM groups. Would it be possible to define the peak of the amyloid aggregation vs baseline curve, and provide separate scatterplots for each side of the peak for annual AV45 change against sTREM2 levels?

Response: Please see our response on the previous comment which addresses this issue. The reviewer is correct, that the effect of sTREM2 on rates of AV45 PET changes is highest at the peak level of the rate of increase in AV45 PET.

8. Comment:

Was Cohen's d calculated by transforming the beta coefficient? For the present analyses, wouldn't variance explained be a more intuitive measure, since relationships between continuous measures are tested, rather than group comparisons?

Response: Cohen's d was computed using the lme.dscore function of the EMAtools package in R (see <https://cran.r-project.org/web/packages/EMAtools/EMAtools.pdf>). Following the reviewers' comment, we now provide partial R^2 values (computed via the r2beta function of the r2glmm package, see <https://cran.r-project.org/web/packages/r2glmm/r2glmm.pdf>) for all reported model terms in table 2.

9. Comment:

What software was used to perform the statistical analyses?

Response: All statistical analyses were performed in R statistical software (Version 3.6.1). We have added this information to the statistics section of the manuscript.

Other questions:

10. Comment:

-the APP NLGF models display microgliosis and astrocytosis, as is also reflected on PET in this study. However, this model does not show neurofibrillary tangles or neuronal loss according to Saito et al., (2014), making it difficult to interpret results as potentially protective effects. How can the argument of protective effects of activated microglia in this mouse model further be supported, other than changes on amyloid PET?

Response: We agree that the investigation of potential protective effects on outcomes other than amyloid deposition is principally of importance. However, A β is a key pathology in AD and this was our primary outcome. The *App*^{NL-G-F} mice examined in the current study shows a relatively slow accumulation of fibrillar A β and plaque morphology similar to humans (Saito et al., 2014). It is true that this mouse model does not develop tau pathology. However, this accounts for most of the commonly used A β mouse models and models that fully recapitulate the full cascade of AD-related brain changes including combined A β , tau pathology and neurodegeneration are not yet established (Jankowsky & Zheng, 2017). We acknowledge that a behavioral characterization of the mice would have been desirable, where the current results encourage such future investigations.

11. Comment: *Figure 2 shows overlap between the wild-type and APP-NLGF mice both for the microglial activation and the rate of change in amyloid PET signals. How can this overlap be interpreted, could this reflect technical issues? Could the wildtype data be added to the scatterplot to show whether these also show a similar relationship as the APP-NLGF mice.*

Response: The reported overlap of TSPO-PET at baseline and of longitudinal A β -PET changes likely represents a mixture between methodological variability and inter-animal heterogeneity. The variability of TSPO- and amyloid-PET signals in wild-type animals in the current study (CoV TSPO-PET baseline: 3.2% CoV amyloid-PET baseline 3.1%) is within the range of variability reported by previous studies (Blume et al., 2018; Brendel, Focke, et al., 2017; Brendel, Jaworska, Herms, Trambauer, Rötzer, et al., 2015; Brendel et al., 2016). Importantly, previous studies demonstrated for both PET tracers a high correlation between PET uptake and the measurements obtained by the corresponding immunohistochemical gold standard methods of amyloid deposition or microglia activity (Brendel, Jaworska, Herms, Trambauer, Rotzer, et al., 2015; Brendel et al., 2016; Parhizkar et al., 2019; Rominger et al., 2013; Sacher et al., 2019), supporting the validity of the PET tracers used in the current study.

In the wild type mice, we find no association between the levels of microglia PET and amyloid PET at baseline ($r=-0.1$, $p=0.534$) nor for microglia PET at baseline as a predictor of the rate of change in amyloid PET ($r=-0.07$, $p=0.655$, see Figure below).

Together, the results are consistent with the interpretation of an abnormal elevation of microglia in response to elevated amyloid PET uptake in the transgenic mouse model.

Given the absence of an association between both tracers and the fact that amyloid PET in the wild type mice represents likely pure noise, we chose not to include that figure in the manuscript.

12. Comment: *Has CSF been collected in these mice or in this model, would it be possible to measure sTREM2 for them as well, to provide more support for the comparison between humans & model (this is not a suggestion to perform if unavailable, in that case perhaps an idea for the future)?*

Response: This is an important point. We have not established CSF extraction in mice in our lab yet. However, we have previously shown an association between TSPO PET levels in amyloid transgenic mice and sTREM2 levels derived from brain homogenate (Brendel, Kleinberger, et al., 2017). However, we agree with the reviewer that any investigation of the correlation between CSF sTREM2 and microglia PET is of high translational interest. Thus, we now mention this point more specifically as a limitation in the discussion to encourage future investigations of that sort (p.20).

13. Comment: *Perhaps the authors could provide more discussion on potential causes of inter-mice differences in microglial activation signal, given that they are genetically identical. One suggestion is given, and discarded (i.e., influence of enhanced environment - but I am assuming these mice were reared in similar environments, which were not necessarily enhanced?)*

Response: High variability of TSPO expression as assessed by PET is not only known for amyloid mouse models and wild-type mice but also for cognitively normal humans (Tuisku et al., 2019). As discussed above, there is a strong correlation between TSPO-PET and immunohistochemistry for activated microglia, supporting the claim to predominantly image heterogeneous microglial activation and not a technical issue. We can confirm that all animals were kept in equal environment at normal housing. Potential reasons for the observed inter-mice differences could be interindividual differences in the level of experienced stress (Calcia

et al., 2016), physical activity (Kohman, Bhattacharya, Wojcik, & Rhodes, 2013) or different stages of the menstrual cycle (Thakkar, Wang, Wang, Vadlamudi, & Brann, 2018). However, these parameters were not assessed in the current study and thus such potential explanations remain to be confirmed.

We added this discussion on p. 23.

References:

- Blume, T., Focke, C., Peters, F., Deussing, M., Albert, N. L., Lindner, S., . . . Brendel, M. (2018). Microglial response to increasing amyloid load saturates with aging: a longitudinal dual tracer in vivo muPET-study. *J Neuroinflammation*, *15*(1), 307. doi:10.1186/s12974-018-1347-6
- Brendel, M., Focke, C., Blume, T., Peters, F., Deussing, M., Probst, F., . . . Rominger, A. (2017). Time Courses of Cortical Glucose Metabolism and Microglial Activity Across the Life Span of Wild-Type Mice: A PET Study. *J Nucl Med*, *58*(12), 1984-1990. doi:10.2967/jnumed.117.195107
- Brendel, M., Jaworska, A., Herms, J., Trambauer, J., Rotzer, C., Gildehaus, F. J., . . . Rominger, A. (2015). Amyloid-PET predicts inhibition of de novo plaque formation upon chronic gamma-secretase modulator treatment. *Mol Psychiatry*, *20*(10), 1179-1187. doi:10.1038/mp.2015.74
- Brendel, M., Jaworska, A., Herms, J., Trambauer, J., Rötzer, C., Gildehaus, F. J., . . . Rominger, A. (2015). Amyloid-PET predicts inhibition of de novo plaque formation upon chronic γ -secretase modulator treatment. *Molecular psychiatry*, *20*(10), 1179-1187. doi:10.1038/mp.2015.74
- Brendel, M., Kleinberger, G., Probst, F., Jaworska, A., Overhoff, F., Blume, T., . . . Rominger, A. (2017). Increase of TREM2 during Aging of an Alzheimer's Disease Mouse Model Is Paralleled by Microglial Activation and Amyloidosis. *Front Aging Neurosci*, *9*, 8. doi:10.3389/fnagi.2017.00008
- Brendel, M., Probst, F., Jaworska, A., Overhoff, F., Korzhova, V., Albert, N. L., . . . Rominger, A. (2016). Glial Activation and Glucose Metabolism in a Transgenic Amyloid Mouse Model: A Triple-Tracer PET Study. *J Nucl Med*, *57*(6), 954-960. doi:10.2967/jnumed.115.167858
- Calcia, M. A., Bonsall, D. R., Bloomfield, P. S., Selvaraj, S., Barichello, T., & Howes, O. D. (2016). Stress and neuroinflammation: a systematic review of the effects of stress on microglia and the implications for mental illness. *Psychopharmacology (Berl)*, *233*(9), 1637-1650. doi:10.1007/s00213-016-4218-9
- Focke, C., Blume, T., Zott, B., Shi, Y., Deussing, M., Peters, F., . . . Brendel, M. (2019). Early and Longitudinal Microglial Activation but Not Amyloid Accumulation Predicts Cognitive Outcome in PS2APP Mice. *J Nucl Med*, *60*(4), 548-554. doi:10.2967/jnumed.118.217703
- Gispert, J. D., Monte, G. C., Falcon, C., Tucholka, A., Rojas, S., Sanchez-Valle, R., . . . Molinuevo, J. L. (2016). CSF YKL-40 and pTau181 are related to different cerebral morphometric patterns in early AD. *Neurobiol Aging*, *38*, 47-55. doi:10.1016/j.neurobiolaging.2015.10.022
- Jankowsky, J. L., & Zheng, H. (2017). Practical considerations for choosing a mouse model of Alzheimer's disease. *Mol Neurodegener*, *12*(1), 89. doi:10.1186/s13024-017-0231-7
- Jay, T. R., Hirsch, A. M., Broihier, M. L., Miller, C. M., Neilson, L. E., Ransohoff, R. M., . . . Landreth, G. E. (2017). Disease Progression-Dependent Effects of TREM2 Deficiency in a Mouse Model of Alzheimer's Disease. *J Neurosci*, *37*(3), 637-647. doi:10.1523/JNEUROSCI.2110-16.2016
- Kohman, R. A., Bhattacharya, T. K., Wojcik, E., & Rhodes, J. S. (2013). Exercise reduces activation of microglia isolated from hippocampus and brain of aged mice. *J Neuroinflammation*, *10*, 114. doi:10.1186/1742-2094-10-114
- Lemoine, L., Leuzy, A., Chiotis, K., Rodriguez-Vieitez, E., & Nordberg, A. (2018). Tau positron emission tomography imaging in tauopathies: The added hurdle of off-target binding. *Alzheimers Dement (Amst)*, *10*, 232-236. doi:10.1016/j.dadm.2018.01.007

- Mattsson-Carlgrén, N., Andersson, E., Janelidze, S., Ossenkoppele, R., Insel, P., Strandberg, O., . . . Hansson, O. (2020). Abeta deposition is associated with increases in soluble and phosphorylated tau that precede a positive Tau PET in Alzheimer's disease. *Sci Adv*, 6(16), eaaz2387. doi:10.1126/sciadv.aaz2387
- Ozmen, L., Albientz, A., Czech, C., & Jacobsen, H. (2009). Expression of transgenic APP mRNA is the key determinant for beta-amyloid deposition in PS2APP transgenic mice. *Neurodegener Dis*, 6(1-2), 29-36. doi:10.1159/000170884
- Parhizkar, S., Arzberger, T., Brendel, M., Kleinberger, G., Deussing, M., Focke, C., . . . Haass, C. (2019). Loss of TREM2 function increases amyloid seeding but reduces plaque-associated ApoE. *Nat Neurosci*, 22(2), 191-204. doi:10.1038/s41593-018-0296-9
- Rominger, A., Brendel, M., Burgold, S., Keppler, K., Baumann, K., Xiong, G., . . . Schlichtiger, J. (2013). Longitudinal assessment of cerebral b-amyloid deposition in mice overexpressing Swedish mutant b-amyloid precursor protein using 18F-florbetaben PET. *J Nucl Med*, 54(7), 1127-1134.
- Sacher, C., Blume, T., Beyer, L., Peters, F., Eckenweber, F., Sgobio, C., . . . Brendel, M. (2019). Longitudinal PET Monitoring of Amyloidosis and Microglial Activation in a Second-Generation Amyloid-beta Mouse Model. *J Nucl Med*, 60(12), 1787-1793. doi:10.2967/jnumed.119.227322
- Saito, T., Matsuba, Y., Mihira, N., Takano, J., Nilsson, P., Itohara, S., . . . Saido, T. C. (2014). Single App knock-in mouse models of Alzheimer's disease. *Nat Neurosci*, 17(5), 661-663. doi:10.1038/nn.3697
- Scholl, M., Lockhart, S. N., Schonhaut, D. R., O'Neil, J. P., Janabi, M., Ossenkoppele, R., . . . Jagust, W. J. (2016). PET Imaging of Tau Deposition in the Aging Human Brain. *Neuron*, 89(5), 971-982. doi:10.1016/j.neuron.2016.01.028
- Suarez-Calvet, M., Kleinberger, G., Araque Caballero, M. A., Brendel, M., Rominger, A., Alcolea, D., . . . Haass, C. (2016). sTREM2 cerebrospinal fluid levels are a potential biomarker for microglia activity in early-stage Alzheimer's disease and associate with neuronal injury markers. *EMBO Mol Med*, 8(5), 466-476. doi:10.15252/emmm.201506123
- Thakkar, R., Wang, R., Wang, J., Vadlamudi, R. K., & Brann, D. W. (2018). 17beta-Estradiol Regulates Microglia Activation and Polarization in the Hippocampus Following Global Cerebral Ischemia. *Oxid Med Cell Longev*, 2018, 4248526. doi:10.1155/2018/4248526
- Tuisku, J., Plaven-Sigray, P., Gaiser, E. C., Airas, L., Al-Abdulrasul, H., Bruck, A., . . . Cervenka, S. (2019). Effects of age, BMI and sex on the glial cell marker TSPO - a multicentre [(11)C]PBR28 HRRT PET study. *Eur J Nucl Med Mol Imaging*, 46(11), 2329-2338. doi:10.1007/s00259-019-04403-7

9th Jun 2020

Dear Prof. Ewers,

Thank you for the submission of your revised manuscript to EMBO Molecular Medicine. We have now received the enclosed report from the two referees who were asked to re-assess it. As you will see the referees are now overall supportive and I am pleased to inform you that we will be able to accept your manuscript pending the following amendments:

1. We would encourage you to address referee #2' comment with regards to adding the scatterplot for WT mice.

2. Figures: figures must be uploaded as single files, separately. For Figure 3, the reference to panels A-C is missing in the main text, please fix.

3. We replaced Supplementary Information with Expanded View (EV) Figures and Tables that are collapsible/expandable online. EV Figures should be cited as 'Figure EV1, Figure EV2" etc... in the text and their respective legends should be included in the main text after the legends of regular figures. See <https://www.embopress.org/page/journal/17574684/authorguide#expandedview>

4. In the main manuscript file, please do the following:

- Please provide up to 5 keywords and incorporate them in the main text
- Please remove the green color font.
- Indicate in legends exact n= and exact p= values, not a range, along with the statistical test used.
- In Materials and Methods, include a statement that the experiments conformed to the principles set out in the WMA Declaration of Helsinki and the Department of Health and Human Services Belmont Report.
- In Materials and Methods, for animal studies, include a statement about randomization even if no randomization was used.

5. Checklist: Both co-corresponding authors' names should be on the checklist.

6. Authors' contribution: MSC, EMR, GK, YD, LP, MAC, LP, CMK, JL contributed but not in author list; Gloria Biechele, Christian Sacher, Tanja Blume, John Q. Trojanowski, Jochen Herms, Martin Dichgans, Matthias Brendl MISSING from contributions.

7. We would also encourage you to include the source data for figure panels that show essential data. Numerical data should be provided as individual .xls or .csv files (including a tab describing the data). For blots or microscopy, uncropped images should be submitted (using a zip archive if multiple images need to be supplied for one panel). Additional information on source data and instruction on how to label the files are available at <https://www.embopress.org/page/journal/17574684/authorguide#sourcedata>

8. Please add a "Data Availability" section after the Materials & Methods and include the following single sentence in this section- "This study includes no data deposited in external repositories".

9. According to our editorial policy with regards to the "conflict of interest" (see below), the current statement suggests that you have no specific financial interest to declare - please confirm that.

'the journal requires authors of original research papers to declare any competing commercial interests in relation to the submitted work. It is difficult to specify a threshold at which a financial interest becomes significant, but as a practical guideline, we would suggest this to be any undeclared interest that could embarrass you were it to become publicly known.'

<https://www.embopress.org/page/journal/17574684/authorguide#conflictsofinterest>

10. For more information: There is space at the end of each article to list relevant web links for further consultation by our readers. Could you identify some relevant ones and provide such information as well? Some examples are patient associations, relevant databases, OMIM/proteins/genes links, author's websites, etc...

11. As part of the EMBO Publications transparent editorial process initiative (see our Editorial at <http://embomolmed.embopress.org/content/2/9/329>), EMBO Molecular Medicine will publish online a Review Process File (RPF) to accompany accepted manuscripts.

In the event of acceptance, this file will be published in conjunction with your paper and will include the anonymous referee reports, your point-by-point response and all pertinent correspondence relating to the manuscript. Let us know whether you agree with this.

12. Please also suggest a striking image or visual abstract to illustrate your article. Please provide such image as a jpeg file 550 px-wide x 400-px high.

13. Please remove the synopsis text from the main manuscript file. I have slightly shortened the Synopsis text. Could you please let me know if it is fine like this or if you would like to introduce further changes?

Synopsis text:

TREM2 is a protein almost exclusively expressed by microglia in the brain. This study investigates the association between the soluble TREM2 (sTREM2) levels in cerebrospinal fluid (CSF) and the longitudinal Ab accumulation in human and mouse.

- In patients with Ab pathology, higher CSF levels of sTREM2 are associated with lower rates of Ab accumulation.
- Higher CSF sTREM2 levels are associated with lower neurofibrillary tangles.
- In the Ab mouse model, higher microglia activation at baseline is associated with lower rates of Ab accumulation between 5 and 10 months of age, when Ab deposition primarily takes place.

I look forward to seeing a revised version of your manuscript as soon as possible.

Sincerely,
Jingyi

Jingyi Hou
Editor
EMBO Molecular Medicine

*** Instructions to submit your revised manuscript ***

To submit your manuscript, please follow this link:

<https://embomolmed.msubmit.net/cgi-bin/main.plex>

- 1) a .docx formatted version of the manuscript text (including Figure legends and tables)
- 2) Separate figure files*
- 3) supplemental information as Expanded View and/or Appendix. Please carefully check the authors guidelines for formatting Expanded view and Appendix figures and tables at <https://www.embopress.org/page/journal/17574684/authorguide#expandedview>
- 4) a letter INCLUDING the reviewer's reports and your detailed responses to their comments (as Word file).
- 5) The paper explained: EMBO Molecular Medicine articles are accompanied by a summary of the articles to emphasize the major findings in the paper and their medical implications for the non-specialist reader. Please provide a draft summary of your article highlighting
 - the medical issue you are addressing,
 - the results obtained and
 - their clinical impact.This may be edited to ensure that readers understand the significance and context of the research. Please refer to any of our published articles for an example.
- 6) For more information: There is space at the end of each article to list relevant web links for further consultation by our readers. Could you identify some relevant ones and provide such information as well? Some examples are patient associations, relevant databases, OMIM/proteins/genes links, author's websites, etc...

7) Author contributions: the contribution of every author must be detailed in a separate section.

8) EMBO Molecular Medicine now requires a complete author checklist (<https://www.embopress.org/page/journal/17574684/authorguide>) to be submitted with all revised manuscripts. Please use the checklist as guideline for the sort of information we need WITHIN the manuscript. The checklist should only be filled with page numbers where the information can be found. This is particularly important for animal reporting, antibody dilutions (missing) and exact values and n that should be indicated instead of a range.

9) Every published paper now includes a 'Synopsis' to further enhance discoverability. Synopses are displayed on the journal webpage and are freely accessible to all readers. They include a short stand first (maximum of 300 characters, including space) as well as 2-5 one sentence bullet points that summarise the paper. Please write the bullet points to summarise the key NEW findings. They should be designed to be complementary to the abstract - i.e. not repeat the same text. We encourage inclusion of key acronyms and quantitative information (maximum of 30 words / bullet point). Please use the passive voice. Please attach these in a separate file or send them by email, we will incorporate them accordingly.

You are also welcome to suggest a striking image or visual abstract to illustrate your article. If you do please provide a jpeg file 550 px-wide x 400-px high.

10) A Conflict of Interest statement should be provided in the main text

11) Please note that we now mandate that all corresponding authors list an ORCID digital identifier. This takes <90 seconds to complete. We encourage all authors to supply an ORCID identifier, which will be linked to their name for unambiguous name identification.

Currently, our records indicate that the ORCID for your account is 0000-0001-5231-1714.

Link Not Available

12) The system will prompt you to fill in your funding and payment information. This will allow Wiley to send you a quote for the article processing charge (APC) in case of acceptance. This quote takes into account any reduction or fee waivers that you may be eligible for. Authors do not need to pay any fees before their manuscript is accepted and transferred to our publisher.

Photos 400-800 DPI

*Additional important information regarding figures and illustrations can be found at <http://bit.ly/EMBOPressFigurePreparationGuideline>

The system will prompt you to fill in your funding and payment information. This will allow Wiley to send you a quote for the article processing charge (APC) in case of acceptance. This quote takes into account any reduction or fee waivers that you may be eligible for. Authors do not need to pay any fees before their manuscript is accepted and transferred to our publisher.

***** Reviewer's comments *****

Referee #1 (Remarks for Author):

The authors have addressed all my comments and included new analyses which have greatly improved the paper. I have no further comments.

Referee #3 (Comments on Novelty/Model System for Author):

I still think that changes over time in abeta accumulation, and the influence of sTREM2 on these changes could be modelled in a more straightforwardly in one linear mixed model. However, the argument to simply take the differences in amyloid between subsequent time points in order to be able to pick up non-linearities is also a valid.

Referee #3 (Remarks for Author):

The authors have clarified all issues, and I have nothing to add, apart from the mini suggestion to reconsider adding the scatterplot for WT mice as well, since this negative finding in my view provides further support that the APP-NLGF correlations are unlikely due to noise.

Editorial Comments

1. Editor: We would encourage you to address referee #2' comment with regards to adding the scatterplot for WT mice.

Response: *The WT mice are now included in the scatterplot in Figure 2C*

2. Editor: Figures: figures must be uploaded as single files, separately. For Figure 3, the reference to panels A-C is missing in the main text, please fix.

Response: *All figures were uploaded as separate files. References to panels A-C of figure 3 have been added to the results section of the manuscript (p.15)*

3. Editor: We replaced Supplementary Information with Expanded View (EV) Figures and Tables that are collapsible/expandable online. EV Figures should be cited as 'Figure EV1, Figure EV2' etc... in the text and their respective legends should be included in the main text after the legends of regular figures. See <https://www.embopress.org/page/journal/17574684/authorguide#expandedview>

Response: *We have changed the label of supplementary figure 1 to Figure EV1, which is referenced on p.14. In line with the journal standards, the respective legend is included in the main text after the legends of the regular figures.*

4. Editor: In the main manuscript file, please do the following:

4.1. Please provide up to 5 keywords and incorporate them in the main text

Response: *Key words are now provided on p.2*

4.2. Please remove the green color font.

Response: *Color fonts have been removed*

4.3. Indicate in legends exact n= and exact p= values, not a range, along with the statistical test used.

Response: *Exact n and p-values have been added to all Figures*

4.4. In Materials and Methods, include a statement that the experiments conformed to the principles set out in the WMA Declaration of Helsinki and the Department of Health and Human Services Belmont Report.

Response: *A statement has been added to the methods section (p.8)*

4.5. In Materials and Methods, for animal studies, include a statement about randomization even if no randomization was used.

Response: *We now state on p.8 that no randomization was used in the current study.*

5. Editor: Checklist: Both co-corresponding authors' names should be on the checklist.

Response: *Both corresponding authors (C.Haass & M.Ewers) are now on the checklist.*

6. Editor: Authors' contribution: MSC, EMR, GK, YD, LP, MAC, LP, CMK, JL contributed but not in author list; Gloria Biechele, Christian Sacher, Tanja Blume, John Q. Trojanowski, Jochen Herms, Martin Dichgans, Matthias Brendl MISSING from contributions.

Response: *The missing authors have been added to the list of contributing authors (p.26).*

7. Editor: We would also encourage you to include the source data for figure panels that show essential data. Numerical data should be provided as individual .xls or .csv files (including a tab describing the data). For blots or microscopy, uncropped images should be submitted (using a zip archive if multiple images need to be supplied for one panel). Additional information on source

data and instruction on how to label the files are available at
<https://www.embopress.org/page/journal/17574684/authorguide#sourcedata>

Response: Source files have now been provided.

8. Editor: Please add a "Data Availability" section after the Materials & Methods and include the following single sentence in this section- "This study includes no data deposited in external repositories".

Response: *A data availability section has been added to p.11.*

9. Editor: According to our editorial policy with regards to the "conflict of interest" (see below), the current statement suggests that you have no specific financial interest to declare - please confirm that.

'the journal requires authors of original research papers to declare any competing commercial interests in relation to the submitted work. It is difficult to specify a threshold at which a financial interest becomes significant, but as a practical guideline, we would suggest this to be any undeclared interest that could embarrass you were it to become publicly known.'

<https://www.embopress.org/page/journal/17574684/authorguide#conflictsofinterest>

Response: *We confirm that we have no specific financial interest to declare. We have added this to p.26.*

10. Editor: For more information: There is space at the end of each article to list relevant web links for further consultation by our readers. Could you identify some relevant ones and provide such information as well? Some examples are patient associations, relevant databases, OMIM/proteins/genes links, author's websites, etc...

Response: *We would be happy if you could include the link to our labs' website and the ADNI database:*

Ewers lab: <https://www.isd-research.de/ewers-lab>

Haass lab: <https://www.biochemie.abi.med.uni-muenchen.de/haass1/index.html>

ADNI: <http://adni.loni.usc.edu>

11. Editor: As part of the EMBO Publications transparent editorial process initiative (see our Editorial at <http://embomolmed.embopress.org/content/2/9/329>), EMBO Molecular Medicine will publish online a Review Process File (RPF) to accompany accepted manuscripts.

In the event of acceptance, this file will be published in conjunction with your paper and will include the anonymous referee reports, your point-by-point response and all pertinent correspondence relating to the manuscript. Let us know whether you agree with this.

Response: *We welcome the transparent editorial process and agree with the journal procedures.*

12. Editor: Please also suggest a striking image or visual abstract to illustrate your article. Please provide such image as a jpeg file 550 px-wide x 400-px high.

Response: *We have uploaded a thumbnail image in 550x400 dimensions together with the manuscript files.*

13. Editor: Please remove the synopsis text from the main manuscript file. I have slightly shortened the Synopsis text. Could you please let me know if it is fine like this or if you would like

to introduce further changes?

Synopsis text:

TREM2 is a protein almost exclusively expressed by microglia in the brain. This study investigates the association between the soluble TREM2 (sTREM2) levels in cerebrospinal fluid (CSF) and the longitudinal A β accumulation in human and mouse.

- In patients with A β pathology, higher CSF levels of sTREM2 are associated with lower rates of Ab accumulation.
- Higher CSF sTREM2 levels are associated with lower neurofibrillary tangles.
- In the A β mouse model, higher microglia activation at baseline is associated with lower rates of A β accumulation between 5 and 10 months of age, when A β deposition primarily takes place.

Response: *We have removed the Synopsis from the main manuscript. Thank you for editing the synopsis, this looks great.*

***** Reviewer's comments *****

Referee #1 (Remarks for Author):

The authors have addressed all my comments and included new analyses which have greatly improved the paper. I have no further comments.

Referee #3 (Comments on Novelty/Model System for Author):

I still think that changes over time in abeta accumulation, and the influence of sTREM2 on these changes could be modelled in a more straightforwardly in one linear mixed model. However, the argument to simply take the differences in amyloid between subsequent time points in order to be able to pick up non-linearities is also a valid.

Referee #3 (Remarks for Author):

The authors have clarified all issues, and I have nothing to add, apart from the mini suggestion to reconsider adding the scatterplot for WT mice as well, since this negative finding in my view provides further support that the APP-NLGF correlations are unlikely due to noise.

Response: *We thank the author for these encouraging remarks. We have added the WT mice data to Figure 2C.*

10th Jul 2020

Please find enclosed the final reports on your manuscript. I am glad to inform you that your manuscript is accepted for publication and is now being sent to our publisher to be included in the next available issue of EMBO Molecular Medicine.

We would like to remind you that as part of the EMBO Publications transparent editorial process initiative, EMBO Molecular Medicine will publish a Review Process File online to accompany accepted manuscripts. If you do NOT want the file to be published or would like to exclude figures, please immediately inform the editorial office via e-mail.

Please read below for additional IMPORTANT information regarding your article, its publication and the production process.

Congratulations on this interesting work,

Jingyi Hou
Editor
EMBO Molecular Medicine

Follow us on Twitter @EmboMolMed
Sign up for eTOCs at embopress.org/alertsfeeds

***** Reviewer's comments *****

*** ** IMPORTANT INFORMATION *** **

SPEED OF PUBLICATION

The journal aims for rapid publication of papers, using the advance online publication "Early View" to expedite the process: A properly copy-edited and formatted version will be published as "Early View" after the proofs have been corrected. Please help the Editors and publisher avoid delays by providing e-mail address(es), telephone and fax numbers at which author(s) can be contacted.

Should you be planning a Press Release on your article, please get in contact with embomolmed@wiley.com as early as possible, in order to coordinate publication and release dates.

LICENSE AND PAYMENT:

All articles published in EMBO Molecular Medicine are fully open access: immediately and freely available to read, download and share.

EMBO Molecular Medicine charges an article processing charge (APC) to cover the publication costs. You, as the corresponding author for this manuscript, should have already received a quote with the article processing fee separately. Please let us know in case this quote has not been received.

Once your article is at Wiley for editorial production you will receive an email from Wiley's Author Services system, which will ask you to log in and will present you with the publication license form for completion. Within the same system the publication fee can be paid by credit card, an invoice, pro forma invoice or purchase order can be requested.

Payment of the publication charge and the signed Open Access Agreement form must be received before the article can be published online.

PROOFS

You will receive the proofs by e-mail approximately 2 weeks after all relevant files have been sent to our Production Office. Please return them within 48 hours and if there should be any problems, please contact the production office at embopressproduction@wiley.com.

Please inform us if there is likely to be any difficulty in reaching you at the above address at that time. Failure to meet our deadlines may result in a delay of publication.

All further communications concerning your paper proofs should quote reference number EMM-2020-12308-V3 and be directed to the production office at embopressproduction@wiley.com.

Thank you,

Jingyi Hou
Editor
EMBO Molecular Medicine

Corresponding author: Michael Ewers, Christian Haass

Journal Submitted to: EMBO Mol Med

Manuscript Number: EMM-2020-12308